# VIP interneurons in mouse primary visual cortex selectively enhance responses to weak but specific stimuli

Daniel J Millman*, Gabriel Koch Ocker, Shiella Caldejon, India Kato, Josh D Larkin, Eric Kenji Lee, Jennifer Luviano, Chelsea Nayan, Thuyanh V Nguyen, Kat North, Sam Seid, Cassandra White, Jerome Lecoq, Clay Reid, Michael A Buice, Saskia EJ de Vries

Allen Institute for Brain Science, Seattle, United States

**Abstract** Vasoactive intestinal peptide-expressing (VIP) interneurons in the cortex regulate feedback inhibition of pyramidal neurons through suppression of somatostatin-expressing (SST) interneurons and, reciprocally, SST neurons inhibit VIP neurons. Although VIP neuron activity in the primary visual cortex (V1) of mouse is highly correlated with locomotion, the relevance of locomotion-related VIP neuron activity to visual coding is not known. Here we show that VIP neurons in mouse V1 respond strongly to low contrast front-to-back motion that is congruent with self-motion during locomotion but are suppressed by other directions and contrasts. VIP and SST neurons have complementary contrast tuning. Layer 2/3 contains a substantially larger population of low contrast preferring pyramidal neurons than deeper layers, and layer 2/3 (but not deeper layer) pyramidal neurons show bias for front-to-back motion specifically at low contrast. Network modeling indicates that VIP-SST mutual antagonism regulates the gain of the cortex to achieve sensitivity to specific weak stimuli without compromising network stability.

*For correspondence:
danielm@alleninstitute.org

Competing interests: The authors declare that no competing interests exist.

## Introduction

Inhibitory interneurons play a major role in establishing the dynamics of cortical microcircuits (*Roux and Buzsáki, 2015*; *Cardin, 2018*). In the superficial layers of the cortex, vasoactive intestinal peptide-expressing (VIP) interneurons regulate feedback inhibition of pyramidal neurons through suppression of Martinotti-type somatostatin-expressing (SST) interneurons (*Pfeffer et al., 2013*). Through this disinhibitory mechanism, VIP interneurons are believed to modulate network dynamics based on the behavioral state of the animal; for instance, VIP neurons in mouse primary visual cortex (V1) are reliably active during periods of locomotion (*Fu et al., 2014*). Moreover, VIP neurons in V1 are a target of top-down inputs and mediate enhancement of local pyramidal cell activity in response to activation of those inputs (*Zhang et al., 2014*). Behaviorally, mouse V1 is necessary for the detection of low contrast visual stimuli (*Glickfeld et al., 2013*), and the optogenetic activation of VIP neurons in mouse V1 lowers contrast detection thresholds whereas the activation of SST or PV neurons raises it (*Cone et al., 2019*). This suggests that the perception of low contrast stimuli is strongly enhanced by VIP neuron activity in V1. Although the activity of VIP neurons has been shown to be suppressed below baseline in response to high contrast full-field grating stimuli of all tested spatial and temporal frequencies (*de Vries et al., 2020*), the responses of VIP neurons to low contrast visual stimuli are not known. To this end, we investigated the influence of stimulus contrast and locomotion on the visual responses of VIP, SST, and pyramidal neurons in mouse V1. SST neurons responded exclusively at high contrast whereas VIP neurons responded exclusively at low contrast with a strong preference for front-to-back motion that is congruent with self-motion during locomotion. As a population, layer 2/3 – but not deeper layer – pyramidal neurons responded more strongly at low

contrast than high contrast and showed a slight, but significant, bias for front-to-back motion. Finally, we made novel extensions of stabilized supralinear network (SSN) models to incorporate the diversity of inhibitory interneuron types and used these models to demonstrate that VIP-driven disinhibition at low contrast can drive large increases in pyramidal neuron activity, despite the relatively low activity of both SST and pyramidal neurons in this contrast regime. The selective enhancement of front-to-back motion could increase the detection of obstacles approaching head-on during locomotion. Based on these results, we conclude that VIP neurons amplify responses of pyramidal neurons to weak but behaviorally-relevant stimuli.

## Results and discussion

We recorded responses to full-field (approximately 120° x 90° of visual space) drifting gratings at eight directions (with a spatial frequency of 0.04 cpd and temporal frequency of 1 Hz) and six contrasts (5–80%) during calcium imaging of mouse Cre lines for *Vip* and *Sst* as well as pyramidal neurons across cortical layers (*Cux2*: layer 2/3; *Rorb*: layer 4; *Rbp4*: layer 5; *Ntsr1*: layer 6) transgenically expressing GCaMP6f (see *Figure 1—source data 1* for numbers of neurons, sessions, and mice in the dataset). The four Cre lines used to image pyramidal neurons were chosen to limit GCaMP expression to neurons in a single layer such that fluorescence contamination from processes (e.g. axons or dendrites) of neurons with somata in different layers is minimized, while providing broad coverage of excitatory neuron types within the target layer (see Materials and methods: Experimental Animals). Although *Cux2* is expressed in both layers 2/3 and 4, it was only imaged in layer 2/3. *Vip* mice were imaged in layer 2/3 where VIP neurons are most abundant (*Tremblay et al., 2016*), whereas *Sst* mice were imaged in layer 4 where SST neurons are most abundant. Notably, most, if not all, SST neurons in layer 4 of V1 are Martinotti cells (*Scala et al., 2019*).

*Figure 1* shows fluorescence traces for four example neurons, of the key Cre lines, as well as stepwise transformations to 'events' in the fluorescence traces and, finally, stimulus-response magnitudes and tuning curves. Events in the fluorescence trace for each neuron were detected using a change-point detection algorithm with an L0-regularization penalty (*de Vries et al., 2020*; *Jewell and Witten, 2018*; *Jewell et al., 2019*). The result is a time series of event onset times and magnitudes proportional to the change in GCaMP fluorescence; individual events likely do not correspond to single action potentials but have a bias toward bursts (*Ledochowitsch et al., 2019*; *Huang et al., 2019*). The response for each trial was computed as the mean event magnitude per second and averaged across trials for each condition.

We observed direction- or orientation-tuned neurons that responded preferentially either to high contrast gratings or low contrast gratings. The majority of neurons were responsive to the stimulus set (*Figure 2a*), measured as a statistically significant bias in responses depending on grating contrast and direction (bootstrapped $\chi^2$ test, p<0.01; see Materials and methods). Substantial differences in contrast and direction tuning were apparent across Cre lines (*Figure 2b–h*). Virtually all VIP neurons responded only at low (<20%) contrast to front-to-back motion (0 degrees; nasal-to-temporal) or an adjacent direction (*Figure 2b*), yielding the greatest direction bias among Cre lines as quantified by the vector sum of direction preferences (*Figure 2c*). The direction of bias was consistent across all *Vip* mice (n = 6 sessions, 3 mice; *Figure 2—figure supplement 1A*) and did not result from stimulus direction-selective running behavior (*Figure 2—figure supplement 1B*). High contrast gratings of all directions significantly suppressed activity in a substantial fraction of VIP neurons whereas such suppression was rare in other Cre lines (*Figure 2d*; *Figure 2—figure supplement 2*). SST neurons had high contrast selectivity, weak direction and orientation selectivity, and varied direction preference (*Figure 2b,e,g,h*), resulting in an average population response that was strong at high contrast across all directions, complementing the non-direction selective suppression at high contrast observed in VIP neurons. Unlike inhibitory interneurons, pyramidal neurons exhibited substantial direction and orientation selectivity and tiled all eight possible direction preferences (*Figure 2b,g,h*).

We statistically-validated the contrast tuning of neurons with a model selection procedure (see Materials and methods: Contrast response function fitting and model comparison) comparing low contrast preference, high contrast preference, and intermediate contrast preference. This analysis confirmed that nearly all VIP neurons were low contrast-preferring and nearly all SST neurons were high contrast-preferring (*Figure 2e*). Contrast preference among pyramidal neurons systematically

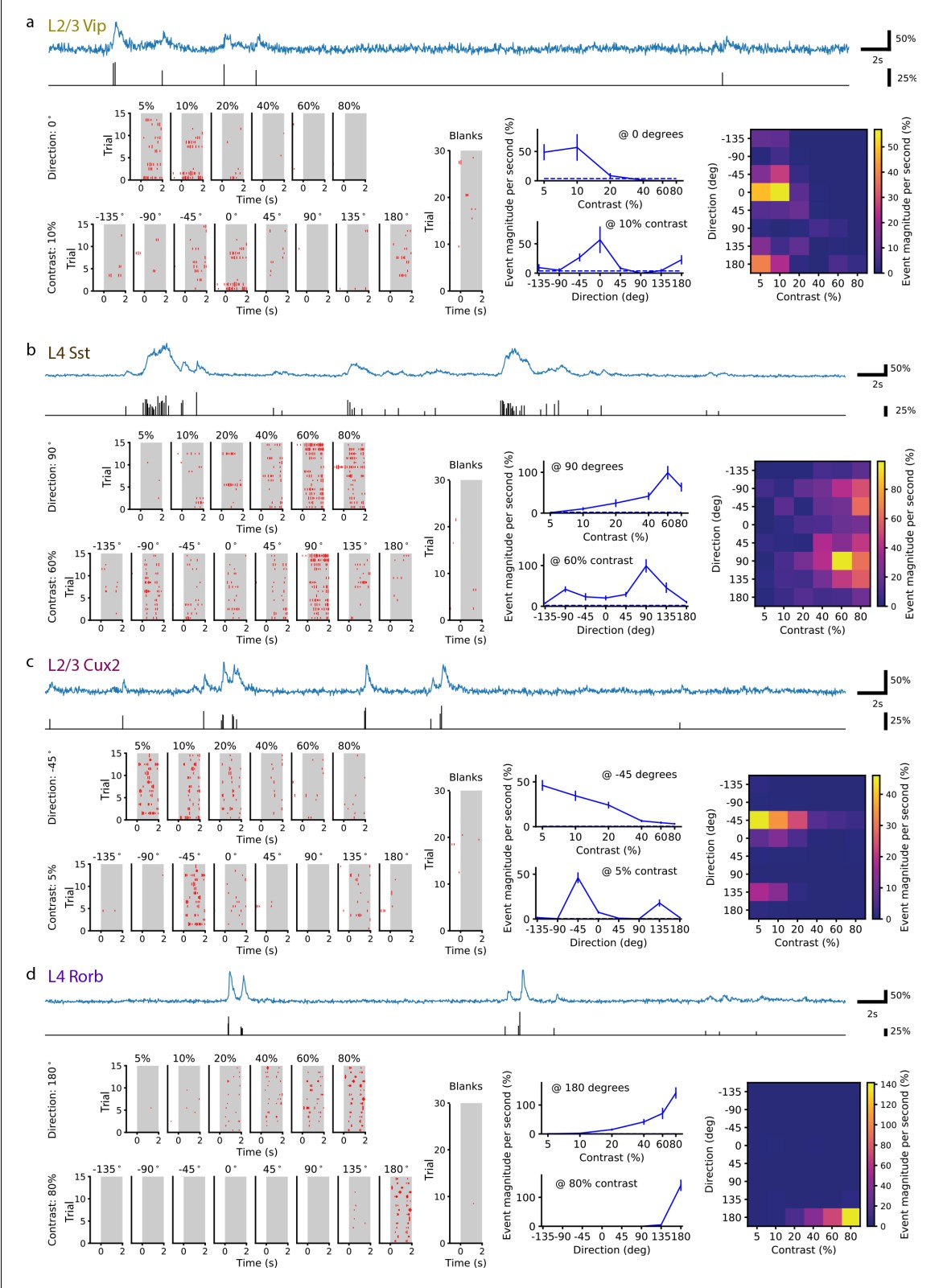

**Figure 1.** Single neurons are tuned for stimulus direction and contrast. (**a**) A single VIP interneuron recorded in layer 2/3 of a Vip mouse responds to low contrast with a preference for motion with a direction of 0 degrees (front-to-back). Top: In blue, 20 s of the dF/F trace for this neuron and, in black, the corresponding events extracted from the dF/F trace. Left: Event rasters for each contrast at the peak direction (0 degrees), each direction at the peak contrast (10%), and blank (i.e. 0% contrast) trials. Middle: Contrast tuning curve at the peak direction and direction tuning curve at the peak

*Figure 1 continued on next page*

*Figure 1 continued*

contrast; mean ± SEM. Right: Heatmap shows the mean response for all stimulus contrasts and directions. (**b**) Same as a, for a single SST neuron recorded in layer 4 of an Sst mouse. This neuron is tuned for high contrast with a preference for motion with a direction of ±90 (up/down). (**c**) Same as a, for a pyramidal neuron recorded in layer 2/3 of a Cux2 mouse. This neuron is tuned for low contrast with a preference for motion with a direction of -45 degrees. (**d**) Same as a, for pyramidal neuron recorded in layer 4 of a Rorb mouse. This neuron is tuned for high contrast with a preference for motion with a direction of 180 degrees (back-to-front).

The online version of this article includes the following source data for figure 1:

**Source data 1.** The total number of cells, experimental sessions, and mice per Cre line.

varied across cortical layers, exhibiting a progression from a mixture of low and high contrast-preferring neurons in layer 2/3 to almost exclusively high contrast-preferring neurons in layers 5 and 6. Like VIP neurons, pyramidal neurons in layer 2/3 showed direction bias toward front-to-back motion at 5% and 10% contrast but not at higher contrasts (*Figure 2c*); pyramidal neurons in deeper layers did not have direction bias. Taken together, concerted changes in response magnitude near 20% contrast across all Cre lines and layers indicate the presence of a phase transition in cortical dynamics between a low contrast regime exemplified by relatively inactive SST neurons and a high contrast regime exemplified by highly active SST neurons.

A previous survey of transcriptomic neuron types using single-cell RNA sequencing identified 16 VIP neuron subtypes, 21 SST neuron subtypes, 3 excitatory neuron subtypes in layer 2/3, 1 excitatory type in layer 4, 12 excitatory types in layer 5, and 17 excitatory types in layer 6 (*Tasic et al., 2018*). That study also investigated the transcriptomic neuron types labeled by the Cre lines used in the present study (see Extended Data Figure 8 of *Tasic et al., 2018*). The *Vip* and *Sst* Cre lines label all transcriptomic subtypes of VIP and SST neurons, respectively, suggesting that common subtypes of SST neurons, such as Martinotti-type SST neurons, are all high contrast-preferring and common subtypes of VIP neurons are all low contrast-preferring. Furthermore, the *Cux2* and *Rorb* Cre lines label all transcriptomic excitatory neuron types in layers 2/3 and 4, respectively. Our finding of substantial populations of both high contrast-preferring and low contrast-preferring neurons in layer 4, where there is only a single transcriptomic excitatory neuron type, demonstrates that neurons of the same transcriptomic subtype can differ in contrast preference. In other layers, whether all neurons of a particular transcriptomic type have the same contrast tuning and, conversely, all neurons with the same contrast tuning correspond to the same transcriptomic type, are important open questions.

Studies of stimulus tuning in the visual system have long reported (*Levick, 1967*; *Rodieck, 1967*) a small but consistent fraction (1–5%) of neurons that exhibit firing rate suppression in response to all stimuli presented, which typically comprised of high contrast gratings, termed 'suppressed-by-contrast' (SbC) neurons. Consistent with a recent report (*de Vries et al., 2020*), these results identify VIP neurons as a major source of SbC neurons in V1. Surprisingly, we observe that not only are these SbC neurons not suppressed at low contrast but that they exhibit robust visual responses to front-to-back motion in such conditions. This contributes new information to our understanding of SbC neurons in the visual circuit. The finding that VIP neurons are suppressed *below* baseline in response to high contrast gratings, rather than suppressed *to* baseline, might be due to the high spontaneous activity of VIP neurons that is available to be suppressed compared to the other neuron types measured here (see Figure 3 as well as Extended Data Figure 1 of *de Vries et al., 2020*). Our measurements of contrast tuning suggest that the high spontaneous activity of VIP neurons enables the cortical circuit to raise or lower the amount of disinhibition of pyramidal neurons depending on stimulus contrast.

To assess the circuit-wide effects of locomotion on cortical dynamics, we examined the average activity of each neuron population as a whole. We focused here on the responses at low contrast in layers 2/3 and 4, but not layers 5 and 6 which did not respond at low contrast. Pyramidal neurons in layers 2/3 and 4, as well as VIP and SST interneurons, had increased activity during stimulus presentations when the mouse was running compared with stimulus presentations when the mouse was stationary (*Figure 3*; *Figure 3—figure supplement 1*). During locomotion, the low contrast and front-to-back direction selectivity that was common to nearly all VIP neurons resulted in an average VIP population response that had tuning closely resembling the tuning of any individual VIP neuron (*Figure 3*, first column). By comparison, the VIP population only weakly responded to front-to-back motion at low contrast when the mice were stationary and did not respond to gratings of any other

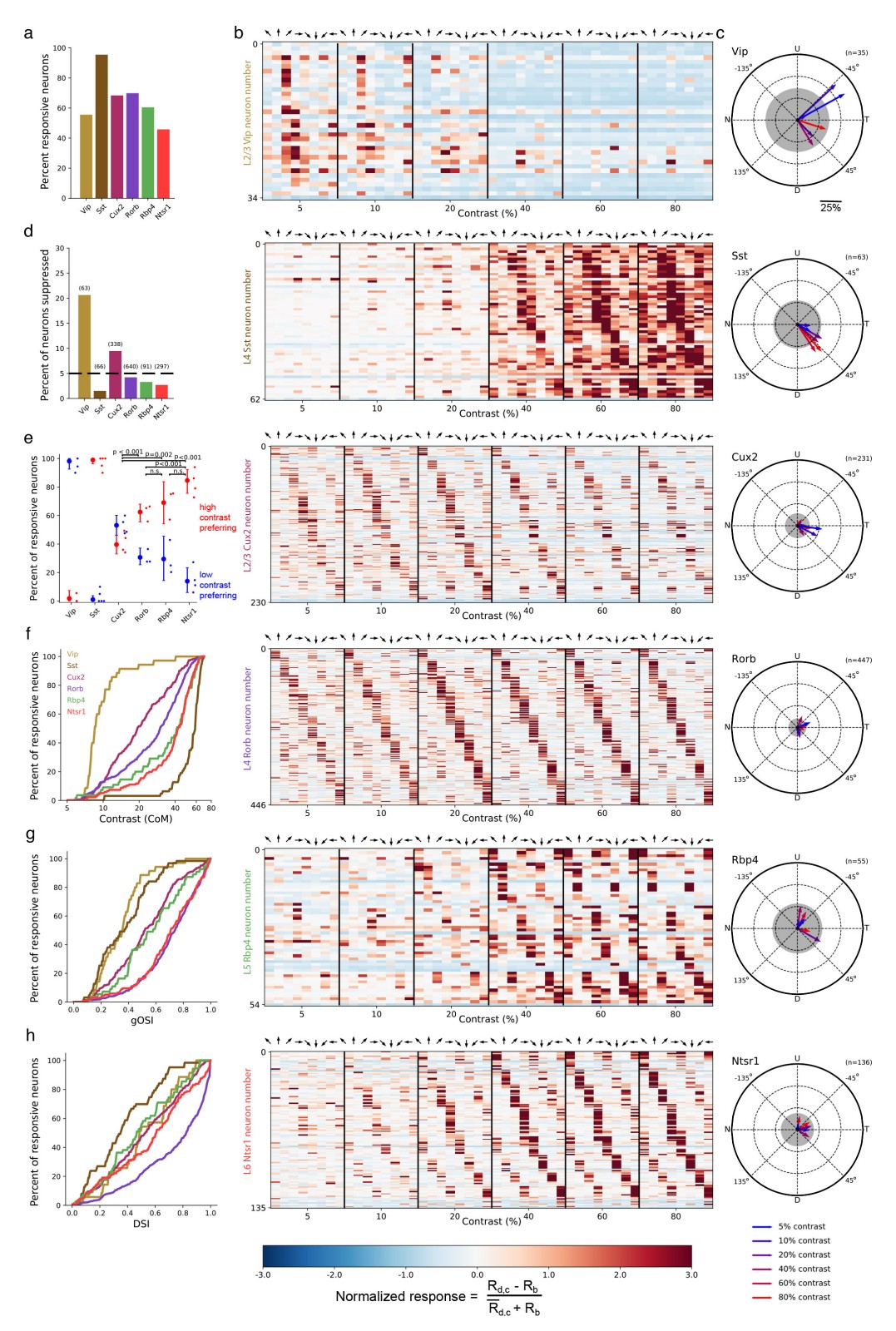

**Figure 2.** Contrast and direction preferences are cell-type and layer specific. (**a**) The fraction of imaged cells that were significantly responsive to the gratings stimulus (bootstrapped $\chi^2$ test, p<0.01). (**b**) Waterfall plots showing the response significance at each contrast and direction of all responsive cells ($\chi^2$ test; p < 0.01) from mice of each Cre line. Each row is one neuron and neurons are ordered by direction preference at the cell's peak contrast. The responses to each stimulus condition are normalized per neuron to be $R_N = (R_{d,c} - R_b)/(\bar{R}_{d,c} + R_b)$, where $R_N$ is the normalized response, $R_{d,c}$ is the

*Figure 2 continued on next page*

*Figure 2 continued*

mean response to a grating with direction *d* and contrast *c*, $R_b$ is the mean blank (0% contrast) response, and $\bar{R}_{d,c}$ is the mean response to gratings across all directions and contrasts. (**c**) Radial plot of the average direction preference of cells of each Cre line at each contrast. Arrows are the vector sum of all responsive cells at a given contrast. Gray shaded region indicates a 95% confidence interval of the vector sum for a population with uniformly-distributed direction preferences, multiple comparisons corrected for the six contrasts. Scale: The distance between each pair of concentric dashed rings is 25%. N: Nasal, T: Temporal, U: Up, D: Down. (**d**) Fraction of all cells of each Cre line that are suppressed by contrast. The mean response to all grating directions at 80% contrast must be significantly below the mean blank response (bootstrapped distribution of mean response differences; family-wise type 1 error < 0.05; see Materials and methods). (**e**) Distribution of contrast response types by Cre line determined by fitting of rising sigmoid (high contrast preferring), falling sigmoid (low contrast preferring), or the product of rising and falling sigmoids (intermediate contrast preferring; not shown due to a very small percentage of neurons tuned for intermediate contrasts). P-values are shown for pairwise comparisons of the fraction of high contrast preferring pyramidal neurons in each layer (bootstrap test of difference of sample proportions). See Materials and methods. (**f**) Cumulative distribution of contrast preferences (center-of-mass of a cell's contrast response function; CoM) across Cre lines. (**g**) Cumulative distribution of global orientation selectivity indices (gOSI) across Cre lines. (**h**) Cumulative distribution of direction selectivity indices across Cre lines.

The online version of this article includes the following figure supplement(s) for figure 2:

**Figure supplement 1.** The direction of VIP neuron bias was consistent across mice and did not result from stimulus direction-selective running behavior.

**Figure supplement 2.** VIP neurons have evoked responses to low contrast gratings but response suppression to high contrast gratings.

direction or contrast. Running also increased the SST population response to high contrast gratings, which also had the highest average response to front-to-back motion but responded strongly as a population to other directions as well (*Figure 3*, second column). The pyramidal population in layer 2/3 (CUX2) responded broadly across directions but more strongly at low than high contrast (*Figure 3*, third column), whereas the pyramidal population in layer 4 (RORB) had comparable response magnitude and running enhancement across contrasts (*Figure 3*, fourth column). This analysis demonstrates a substantial enhancement of responses to low contrast visual stimuli during locomotion that is specific to layer 2/3 pyramidal neurons and VIP neurons.

We built a Generalized Linear Model of VIP, SST, and layer 2/3 pyramidal neuron responses to investigate the contribution of stimulus contrast, stimulus direction, locomotion, and the interactions between these terms to the average activity of each neuron population using a Poisson model to predict responses (*Figure 4a*). To identify only the terms that significantly contribute to activity, we included an L1-regularization penalty in the cost function which resulted in relatively few non-zero terms (12–15 non-zero out of 126 total terms). VIP neurons had the highest weights for blank sweep, low contrasts (5–20%), running, directions of 0° and 180°, running by direction interactions at 0° and 45°, and direction by contrast interactions at ±45° and low contrasts (*Figure 4b*). SST neurons had the highest weights for high contrasts (40–80%), direction of 0°, and running by direction interactions at all directions (*Figure 4c*). Layer 2/3 pyramidal neurons have significant weights only for running, low contrasts (5–20%), and all directions (*Figure 4d*). Overall, this analysis confirms the influence of running, stimulus direction, and stimulus contrast but suggests that interactions among these variables is limited.

Anatomical and optogenetic perturbation experiments suggest that VIP neurons disinhibit pyramidal neurons through their inhibition of SST neurons (*Pfeffer et al., 2013*; *Zhang et al., 2014*; *Pi et al., 2013*). However, VIP neurons only respond to one direction of low contrast grating and SST neurons have very weak responses to low contrast gratings of any direction, potentially limiting the magnitude of SST activity that is available to be inhibited by VIP neurons and, consequently, limiting the magnitude of disinhibition of pyramidal neurons. Evidence that visual cortex has higher gain at low contrast than high contrast (*Heuer and Britten, 2002*; *Cavanaugh et al., 2002*; *Carandini and Heeger, 2012*) suggests that a small reduction in feedback inhibition (e.g. disinhibition) is capable of driving a large increase in pyramidal neuron activity (*Hertäg and Sprekeler, 2019*). We hypothesized that VIP neurons are essential to establishing the high gain regime at low contrast as a result of VIP-mediated disinhibition forming a positive feedback loop (i.e. Pyr → VIP → SST → Pyr) that depends upon, and contributes to, network dynamics. Stabilized supralinear network (SSN) models have been proposed to account for a variety of contrast-dependent response properties in visual cortex (*Rubin et al., 2015*; *Ahmadian et al., 2013*), including the transition from a high gain regime at low contrast to a feedback inhibition dominated low gain regime at high contrast (*Adesnik, 2017*; *Sanzeni et al., 2020*), as well as cortical noise correlations (*Hennequin et al.,*

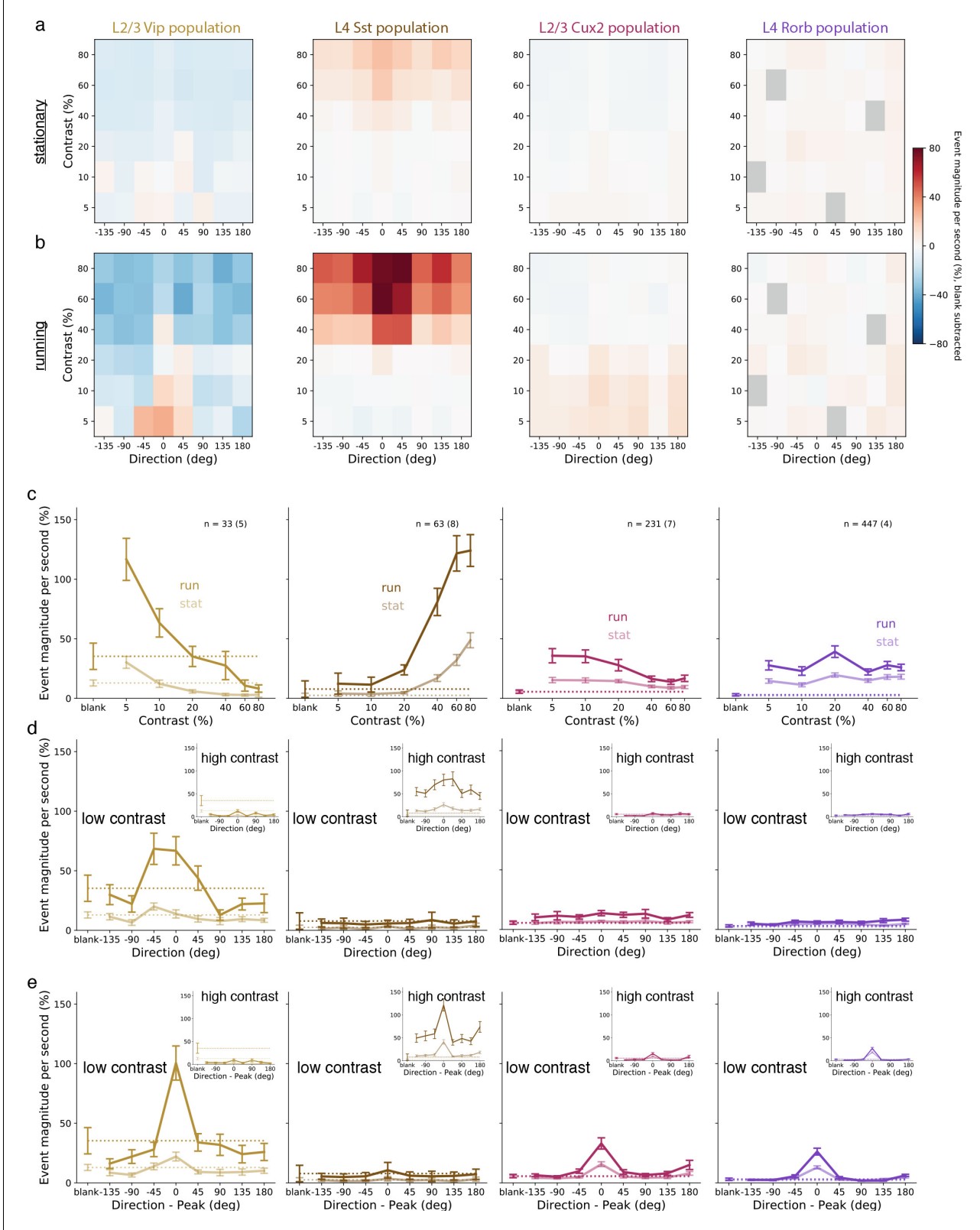

**Figure 3.** Average population responses of inhibitory, but not excitatory, cells are strongly biased toward front-to-back visual motion which is enhanced during locomotion. (a) Mean blank-subtracted event magnitude (a.u.; extracted events derived from dF/F trace) of all neurons from mice of each superficial Cre line during stationary periods. Gray boxes in Rorb plots indicate insufficient run and stationary data. (b) Same as a, for running periods. (c) Mean population contrast responses tuning at peak direction during stationary (faint lines) and running (bold lines) periods. (d) Mean population

*Figure 3 continued on next page*

*Figure 3 continued*

direction response tuning at low (5-10%) contrast. Insets: mean population direction response tuning at high (60-80%) contrast. (**e**) Mean single-neuron direction tuning (i.e. aligned to each neuron's peak direction). Insets: mean single-neuron direction tuning at high (60-80%) contrast. All error bars are SEM. Sample size indicates number of neurons with number of experiments in parenthesis.

The online version of this article includes the following figure supplement(s) for figure 3:

**Figure supplement 1.** Distributions of single neuron response magnitudes across stimulus conditions for key Cre lines.

2018), surround suppression (*Liu et al., 2018*), and effects of feature and spatial attention on neural activity (*Lindsay et al., 2020*). In SSNs, high gain is achieved through supralinear single-neuron transfer functions (e.g. f-I curve) and strong recurrent excitatory connections but the gain is eventually reduced as external input strength increases due to the recruitment of inhibitory neurons which also have supralinear transfer functions (*Miller and Troyer, 2002*; *Priebe and Ferster, 2008*; *Margrie et al., 2002*; *Linaro et al., 2019*). The ability of SSNs to account for a wide variety of phenomenology by utilizing only a few simplified but universal features of cortical circuits (e.g. recurrent excitation, feedback inhibition, and supralinear f-I curves) has established them as attractive models for explaining cortical dynamics (*Kraynyukova and Tchumatchenko, 2018*). However, the impact of interneuron diversity on the behavior of SSNs is largely unknown.

To investigate the distinct roles of each interneuron type, we extended the SSN model from one homogeneous population of interneurons to three populations corresponding to VIP, SST, and par-valbumin-expressing (PV) neurons to model layer 2/3 of mouse V1 (*Figure 5a*; see Materials and methods for further details). Briefly, the network is a ring model in which each layer 2/3 pyramidal neuron ('CUX2') receives external ('sensory') excitatory input that has Gaussian tuning with mean (i.e. peak/preferred direction) corresponding to the neuron's position on the ring and standard deviation of 30 degrees; PV neurons also receive external input which is not tuned (*Kerlin et al., 2010*). SST neurons do not receive external input in our model to incorporate the finding of weak or no thalamocortical input to SST neurons in mouse somatosensory cortex (*Cruikshank et al., 2010*). VIP neurons also do not receive external input in our model since experimental measurements of this input to VIP neurons are lacking, and eliminating this potential source of excitatory input to VIP neurons is the most conservative assumption for reproducing the strong responses of VIP neurons to weak stimuli. The strength of external input is intended to represent a monotonically-increasing function of stimulus contrast, though no specific relationship is claimed here. Connections from CUX2 neurons (i.e. excitatory connections) also have Gaussian tuning that depends on the difference between the orientation preferences of the pre- and post-synaptic neurons (*Figure 5b* top), with broader tuning of connections targeting SST and PV neurons (standard deviation of 100 degrees) than those targeting pyramidal and VIP neurons (standard deviation of 30 degrees) to reflect the relative tuning of the postsynaptic neurons types (*de Vries et al., 2020*; *Kerlin et al., 2010*). Connections from inhibitory neurons (i.e. inhibitory connections) were broadly tuned as well (standard deviation of 100 degrees; *Figure 5b* bottom). To incorporate the bias we measured in direction tuning, we included a minor (~2%) over-representation of pyramidal neurons that prefer the zero degrees direction as well as a bias in the direction tuning of external input for zero degrees to account for the known direction bias of thalamocortical inputs (*Marshel et al., 2012*; *Zhang et al., 2020*). All neurons are modeled as rate units with rectified quadratic transfer function, the simplest supralinear polynomial.

This model is able to qualitatively reproduce the population direction and contrast tuning we observed for VIP, SST, and layer 2/3 pyramidal (CUX2) neurons as well as make a prediction for the tuning of PV neurons (*Figure 5c*). Model VIP neurons are suppressed at high levels of external input regardless of stimulus direction, reproducing the suppressed-by-contrast behavior we observed in our imaging experiments, and active for all stimulus direction at low contrast but most active for the zero degrees direction, again reproducing VIP neuron tuning (*Figure 3b*). The external input strength for which VIP neurons are most active (~10 a.u.) corresponds to the highest gain ('supralinear') regime for L2/3 pyramidal and PV activity (*Figure 5d*: left). Ablating the VIP-to-SST inhibitory connection, the only output of VIP neurons contained in the model, results in a large reduction in the gain and activity of VIP, L2/3 pyramidal, and PV populations at low input (*Figure 5d*: right). Even in the absence of inhibition from VIP neurons, SST neurons have relatively low activity at a low level of external input, demonstrating that suppression of a relatively small amount of SST neuron activity

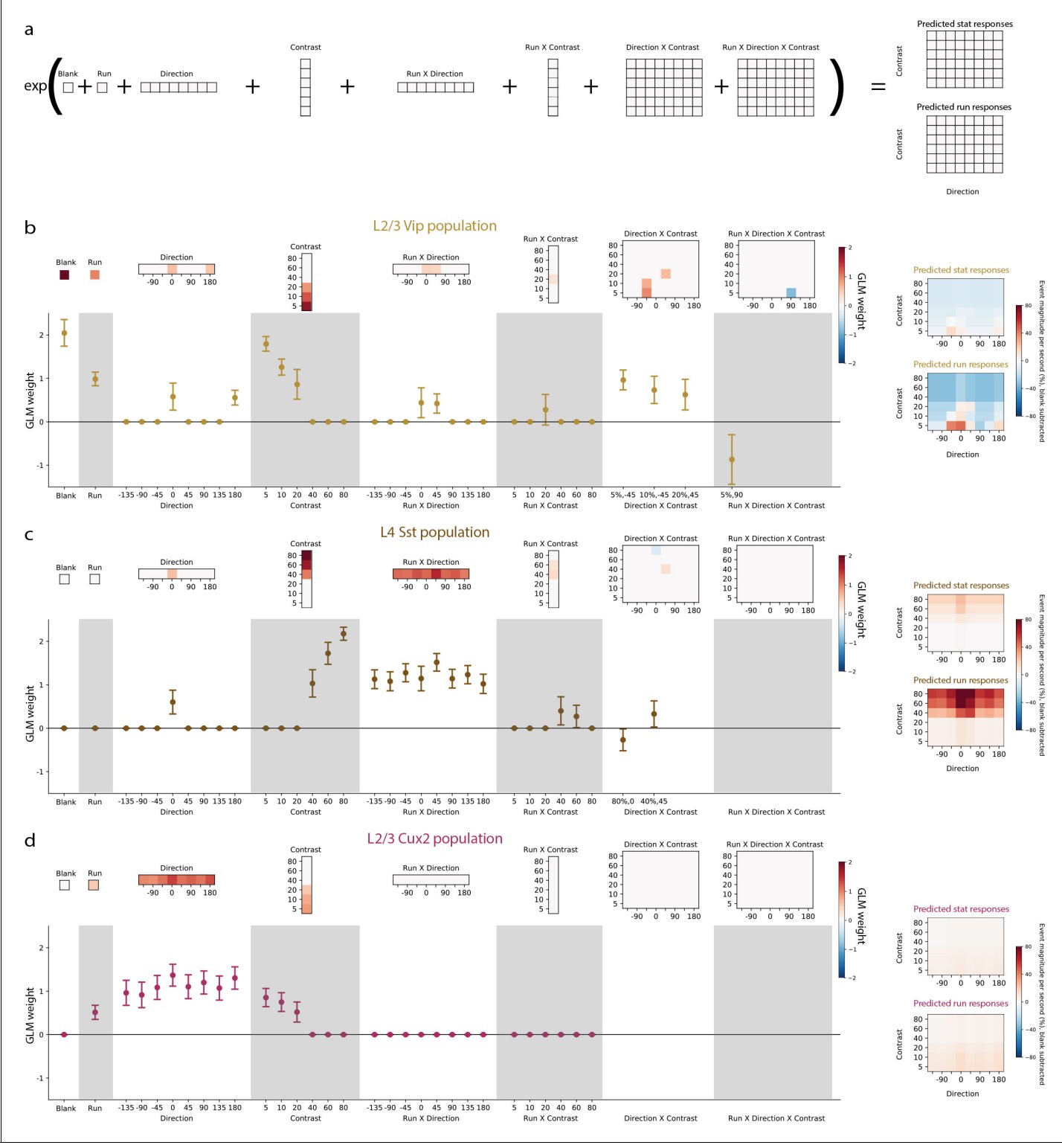

**Figure 4.** Generalized Linear Models reveal the contribution of stimulus direction, stimulus contrast, locomotion, and the interactions between these terms, to the activity of neuronal populations. (a) Schematic of the Poisson GLM consisting of a blank term, a binary run state term (1 for running, 0 for stationary), 8 direction terms, 6 contrast terms, 8 run x direction interaction terms, 6 run x contrast interaction terms, 48 direction x contrast interaction terms, and 48 run x direction x contrast interaction terms. The responses are predicted by summing these 126 terms and raising the sum to an exponential. (b) GLM results for the population of layer 2/3 VIP neurons recorded from Vip mice. Left: The model weights are shown as heatmaps (top) as well as means and 95% confidence intervals (bottom). Sparse weights were obtained using an L1-regularization penalty, resulting in the majority of

*Figure 4 continued on next page*

Figure 4 continued

weights to be zero. For direction x contrast and run x direction x contrast interaction terms, means and confidence intervals are only shown for terms with non-zero weights. Right: Predicted responses to stimulus conditions minus predicted blank response when the mouse is stationary (top) and running (bottom). (c) Same as b, but for the population of layer 4 SST neurons recorded from Sst mice. (d) Same as b, but for the population of layer 2/3 pyramidal neurons recorded from Cux2 mice.

can drive large increases in pyramidal neuron gain. These results indicate that VIP-mediated disinhibition is capable of producing substantial increases in gain at weak inputs, despite low activity of the intermediate SST neuron population, in networks with supralinear single neuron transfer functions and recurrent excitation.

The introduction of a positive feedback loop into the SSN model in the form of VIP-mediated disinhibition could have a destabilizing effect on network dynamics. A key aspect of the stability of network dynamics is whether recurrent excitation required feedback inhibition to prevent runaway activity; that is, whether the network is inhibitory stabilized ('ISN') or non-inhibitory stabilized ('non-ISN'). In this context, achieving a high gain might push networks to the brink of instability. Conversely, suppressing VIP neuron activity *below* baseline, rather than *to* baseline, for high external input strength could be an important component of ensuring network stability. We assessed the stabilization regime by a linear stability analysis of the network's response to a perturbation of the inputs that uniformly targeted all locations on the ring (Materials and methods). The excitatory-excitatory (E-E) component of the network's linear response matrix exposes inhibitory stabilization: if it is negative, the excitatory-excitatory subnetwork has weak effective coupling and does not require inhibition to stabilize it. On the other hand, if the E-E subnetwork's linear response is positive, but the network as a whole is stable (e.g. converges to stable steady-state rates), inhibition is required to stabilize the recurrent excitation and the network is inhibitory-stabilized, an ISN. We find that the network transitions from non-ISN to ISN as external input strength increases with a transition between stability regimes around an external input strength of 50 a.u. (*Figure 5g*), as long as the VIP-to-SST connection strength is below a critical value (normalized to be 1.0 in *Figure 5g–i*). Above the critical VIP-to-SST connection strength, the network becomes highly unstable even for low external input strengths and firing rates of pyramidal, PV, and VIP neurons explode while SST neuron activity is suppressed (*Figure 5h*). To determine the impact of VIP neurons on circuit gain, we measured the pyramidal neuron gain (i.e. the slope with respect to external input strength in *Figure 5h*) and examined the difference between that gain and the gain in a network with VIP-to-SST connection strength set to zero (but otherwise identical). This difference is greatest at the same low external input strengths for which VIP neuron activity is high and monotonically increases as a function of the VIP-to-SST connection strength until the transition to instability at the critical value (*Figure 5i*). The gain enhancement is present only at low external input strengths for which the network is in a non-ISN regime, ensuring that the impact of VIP neurons on gain does not disrupt inhibitory stabilization. Still, the external input strength at which VIP rates explode *above* the critical VIP-to-SST connection strength (*Figure 5h*) closely matches the external input strength at which VIP rates and network gain are the highest *below* the critical VIP-to-SST connection strength (*Figure 5i*), emphasizing the delicate balance of gain and stability in the cortical network. Although VIP-mediated disinhibition and increase in network gain can be destabilizing above a critical strength, this analysis demonstrates that a substantial increase in gain can be achieved over a wide range of VIP-to-SST strengths.

Finally, we also investigated the impact of the relative strength of the weight of PV inputs versus SST inputs to pyramidal neurons ($W_{PV}/(W_{PV}+W_{SST})$; *Figure 5j–l*). Along with the strength of the VIP-to-SST connection, this ratio is a key determinant of how inhibition is recruited in the SSN. Although the external input strength at which the network transitions from non-ISN to ISN decreases as the relative weight from PV neurons increase, the stability behavior (*Figure 5j*), firing rates (*Figure 5k*), and gain effect (*Figure 5l*) remain similar over a broad range of the $W_{PV}/(W_{PV}+W_{SST})$ ratio. Only when input from PV neurons greatly outweighs input from SST neurons, a bifurcation occurs (near $W_{PV}/(W_{PV}+W_{SST})=0.8$) and the network ultimately becomes unstable (near $W_{PV}/(W_{PV}+W_{SST})=0.95$). These results demonstrate that the stability behavior and gain effects we observe in SSNs with three interneuron populations are robust over a wide range of values for key model parameters.

This survey of contrast tuning in mouse V1 revealed two distinct regimes of cortical dynamics in superficial layers of cortex. At high contrast, SST neuron activity is high, VIP neuron activity is

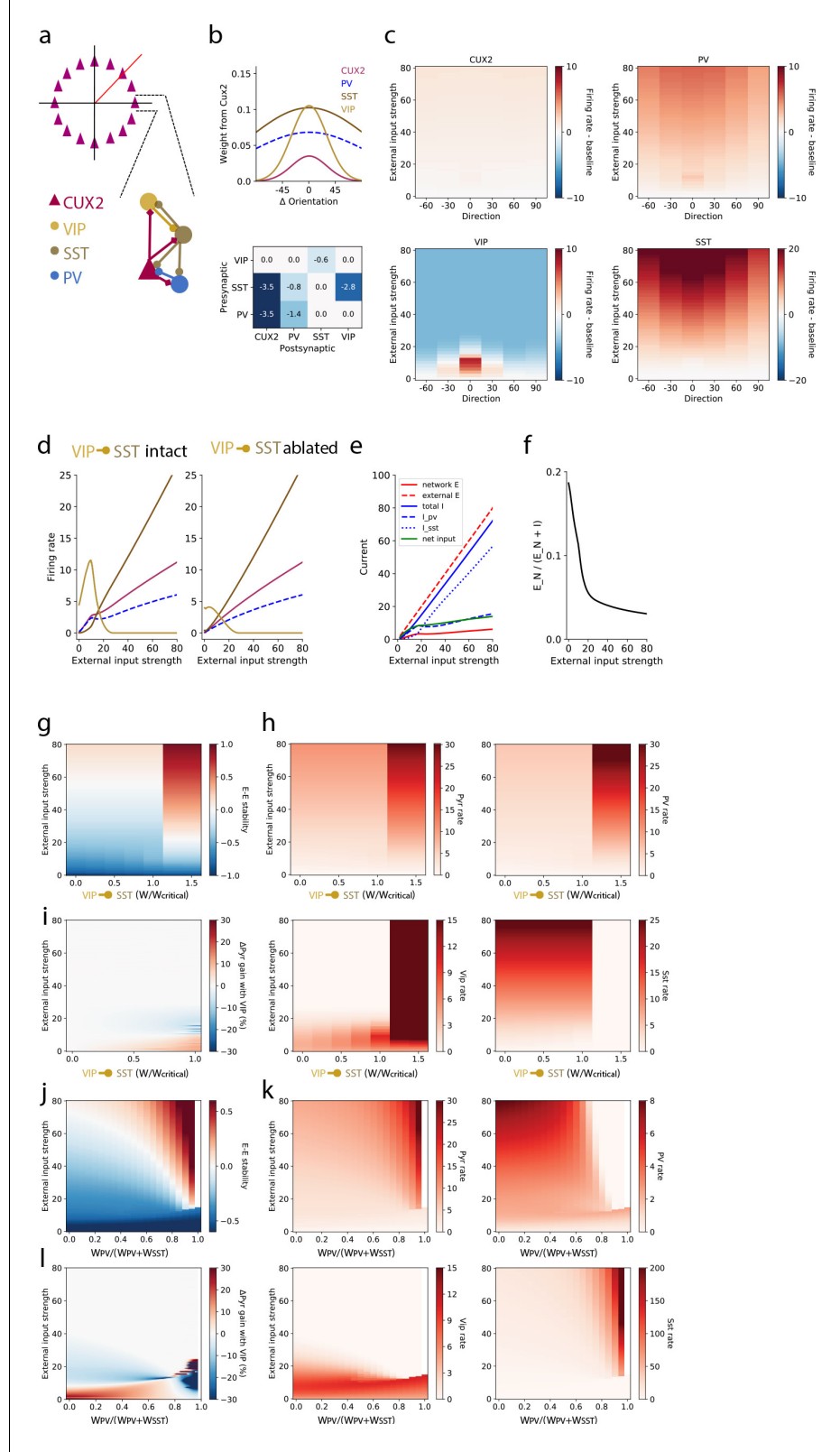

**Figure 5.** A stabilized supralinear network (SSN) model with three interneuron populations reproduces contrast and direction tuning of multiple neuron types and implicates VIP neurons in enhancement of network gain for weak inputs. (a) Top: The network architecture is a ring corresponding to the peak of each L2/3 pyramidal ("CUX2") neuron's direction tuning curve. The entire ring spans 180 degrees of direction. Bottom: A schematic

*Figure 5 continued on next page*

*Figure 5 continued*

illustrates the connectivity among neuron types. (**b**) Top: The distribution of excitatory connection strength from CUX2 pyramidal neurons onto each neuron type is Gaussian with mean equal to the difference in orientation preference of pre- and post-synaptic neurons. The distributions of recurrent connections onto CUX2 neurons and connections onto VIP neurons are narrow (standard deviation of 30 degrees) compared to the distributions onto PV and SST neurons (standard deviation of 100 degrees). Bottom: Inhibitory connection weights are all broadly tuned (standard deviation of 100 degrees). (**c**) The average population responses across direction and contrast conditions qualitatively reproduce experimental data for CUX2, SST, and VIP neurons shown in *Figure 3*. (**d**) Left: The steady state firing rates are shown for model neurons of each type with peak direction tuning of zero degrees in response to an external input of zero degrees. Right: The steady state firing rates of the same model neurons in response to an external input of zero degrees with the VIP-to-SST connection strength set to zero demonstrates that this connection is necessary for a high gain of CUX2 and PV neurons at the low input levels for which VIP neurons are most responsive. (**e**) Currents to the pyramidal neurons in panel d show that most additional external excitatory input above 15 is offset by the recruitment of inhibition. Inhibition from PV neurons dominates at weak external input strengths while inhibition from SST neurons dominates at strong external input strengths. (**f**) The relative fraction of currents that pyramidal neurons receive from other pyramidal neurons, rather than inhibitory neurons, decreases as external input strength increases. This shows the relative dominance of inhibition over excitation in the network. (**g**) The linear stability of the E-E subnetwork shows a transition from non-ISN dynamics (E-E stability < 0) to ISN dynamics (E-E stability > 0) at an external input strength of ~55 for networks with connection weight from VIP to SST neurons below a critical value ($W_{VIP \rightarrow SST} \approx -0.6$). Above the critical value, the E-E subnetwork is highly unstable for all external input strengths greater than ~20. (**h**) The firing rates of all four neuron types remain similar below the critical value of $W_{VIP \rightarrow SST}$, except VIP neurons which increase substantially with increasing $W_{VIP \rightarrow SST}$ but remain most active at weak external input strengths. Above the critical value, rates of pyramidal, PV and VIP neurons increase substantially, and SST neuron rates are near zero, for all external input strengths. (**i**) The effect of the VIP-to-SST neuron connection on pyramidal neuron gain shows that the increase in gain occurs only at weak external input strengths. The gain effect increases with increasing $W_{VIP \rightarrow SST}$ below the critical value. W/Wcritical for the networks shown in panels b-f is 0.99, where W is -0.6. (**j-l**) Same as (**g-i**), except varying the weight of PV inputs onto pyramidal neurons relative to the total weight of inhibitory (i.e. PV and SST) inputs onto pyramidal neurons. The external input strength at which the E-E stability (panel **j**) transitions from non-ISN to ISN decreases as the relative weight from PV neurons increase, but the stability behavior, firing rates (panel **k**), and gain effect (panel **l**) remain the similar until a bifurcation near $W_{PV}/(W_{PV}+W_{SST})=0.8$. The network becomes unstable at very high relative PV weights ($W_{PV}/(W_{PV}+W_{SST}) > 0.95$). $W_{PV}/(W_{PV}+W_{SST})$ for the networks shown in panels b-f is 0.5.

suppressed, and layer 2/3 pyramidal neuron activity is lower than it is at low contrast; at low contrast, SST neuron activity is low, VIP neuron activity is direction tuned and gated by locomotion, and layer 2/3 pyramidal neuron activity is higher and more enhanced by locomotion. Measurements of size tuning with high contrast gratings have shown that SST neurons prefer large gratings, suggestive of a role mediating surround suppression, whereas VIP neurons only respond to gratings smaller than those that drive SST neurons (*Adesnik et al., 2012*; *Dipoppa et al., 2018*). Interestingly, the receptive fields of VIP neurons are larger than those of SST or pyramidal neurons when measured with sparse noise stimuli (*de Vries et al., 2020*), indicating that the selectivity of VIP neurons for small stimuli does not arise simply from having small linear receptive fields. This complementary size tuning parallels the complementary contrast tuning observed here, suggesting that VIP and SST neurons in V1 are tuned for weak and strong inputs, respectively, across multiple stimulus dimensions. Indeed, this relationship appears to hold across sensory modalities as VIP neurons in mouse primary auditory cortex are selective for lower sound intensities than SST or PV neurons (*Mesik et al., 2015*). Taken together, a parsimonious explanation of these results is that VIP neuron activity supports a high gain regime that increases sensitivity to weak inputs, whereas SST neuron activity promotes a low gain regime that decreases sensitivity to strong inputs and maintains network stability. Heightened sensitivity to detect low contrast objects or obstacles approaching head-on during locomotion might be more behaviorally relevant than other directions of motion. This ability of VIP neurons to promote a high gain in the local microcircuit might be indicative of a more general role at the nexus of top-down (e.g. attention) and bottom-up (e.g. saliency) processes.

# Materials and methods

**Key resources table**

| Reagent type (species) or resource | Designation | Source or reference | Identifiers | Additional information |
|---|---|---|---|---|
| Genetic reagent (*M. musculus*) | *Vip*-IRES-Cre | Jackson Laboratory | Stock #: 010908; RRID:MGI:4436915 | Dr. Z Josh Huang (Cold Spring Harbor Laboratory) |
| Genetic reagent (*M. musculus*) | *Sst*-IRES-Cre | Jackson Laboratory | Stock #: 013044; RRID:IMSR_JAX:013044 | Dr. Z Josh Huang (Cold Spring Harbor Laboratory) |
| Genetic reagent (*M. musculus*) | *Cux2*-CreERT2 | MMRRC | RRID:MMRRC_032779-MU | PMID:22879516 |
| Genetic reagent (*M. musculus*) | *Rorb*-IRES2-Cre | Jackson Laboratory | Stock #: 023526 RRID:IMSR_JAX:023526 | PMID:25071457 |
| Genetic reagent (*M. musculus*) | *Rbp4*-Cre_KL100 | MMRRC | RRID:MMRRC_031125-UCD | PMID:24360541 |
| Genetic reagent (*M. musculus*) | *Ntsr1*-Cre_GN220 | Jackson Laboratory | Stock #: 017266; RRID:MMRRC_030648-UCD | PMID:24360541 |
| Genetic reagent (*M. musculus*) | *CaMKII*-tTA x Ai93-GCaMP6f | Jackson Laboratory | Stock #: 024108; RRID:IMSR_JAX:024108 | PMID:22855807; PMID:25741722 |
| Genetic reagent (*M. musculus*) | Ai148-GCaMP6f | Jackson Laboratory | Stock #: 030328; RRID:IMSR_JAX:030328 | PMID:30007418 |
| Software, algorithm | NumPy | NumPy | RRID:SCR_008633 | |
| Software, algorithm | Matplotlib | MatPlotLib | RRID:SCR_008624 | |
| Software, algorithm | pandas | pandas | DOI:10.5281/zenodo.3509134 | |
| Software, algorithm | statsmodel | statsmodel | RRID:SCR_016074 | |
| Software, algorithm | scipy | SciPy | RRID:SCR_008058 | |
| Software, algorithm | scikit-learn | scikit-learn | RRID:SCR_002577 | |

## Experimental animals

All animal procedures were approved by the Institutional Animal Care and Use Committee (IACUC) at the Allen Institute for Brain Science. Six double or triple transgenic mouse lines were used to drive expression of GCamp6/f in genetically-defined cell types, including four excitatory (*Cux2*-CreERT2; *Camk2a*-tTA;Ai93, *Rorb*-IRES2-Cre;*Camk2a*-tTA;Ai93, *Rbp4*-Cre_KL100;*Camk2a*-tTA;Ai93, and *Ntsr1*-Cre_GN220;Ai148) and two inhibitory (*Vip*-IRES-Cre;Ai148 and *Sst*-IRES-Cre;Ai148) mouse lines. Mice were habituated to head fixation and visual stimulus presentation for 2 weeks before data collection. Post-surgical experimental mice were housed in cages individually and maintained on a reverse dark-light cycle with experiments conducted during the dark phase. (See *de Vries et al., 2020* for further Cre line, surgical, and habituation details). Sample size was determined qualitatively to balance repeated experiments for each layer/Cre-line and to preserve the breadth of the survey.

The correspondence between Cre lines (including all six Cre lines used in this study) and transcriptomic neuron subtypes as measured with single-cell RNA sequencing has been reported in Extended Figure 8 of *Tasic et al., 2018*. *Vip*-Cre and *Sst*-Cre lines provide broad coverage of VIP and SST neuron transcriptomic subtypes (16 and 21 subtypes, respectively). In layer 2/3, *Cux2*-CreERT2 labels all three excitatory neuron transcriptomic subtypes. Layer 4 contains only a single transcriptomic neuron type, which is sampled by the *Rorb*-Cre line. *Rbp4*-Cre_KL100 labels all twelve layer 5 neuron transcriptomic subtypes; note that layer 5 was imaged at a single depth in this study, which might result

in sampling only a subset of the layer 5 transcriptomic types. *Ntsr1*-Cre labels all six layer 6 cortico-thalamic neuron transcriptomic subtypes.

## Two-photon imaging platform and image processing

Data was collected using the same data collection pipeline as the Allen Brain Observatory and processed using the same image processing and event detection methods (See *de Vries et al., 2020* for further imaging and image processing details). Calcium imaging was performed with Nikon A1R MP + two-photon microscopes adapted to provide space to accommodate the running disc. Laser excitation with a wavelength of 910 nm was provided by a Ti:Sapphire laser (Chameleon Vision—Coherent). Precompensation was fixed at 10,000 fs$^2$. Movies were recorded at 30 Hz with resonant scanners over a 400 µm field of view with a resolution of 512 × 512 pixels. Temporal synchronization of calcium imaging, visual stimulation, and running wheel movement was achieved by recording all experimental clocks on a single NI PCI-6612 digital IO board at 100 kHz. PMT gain and laser power were chosen for each experiment to maximize dynamic range while saturating fewer than 1000 pixels in the field of view. Two z-stacks, one local (±30 µm from imaging depth in 0.1 µm steps) and one full-depth of the cortex (~700 µm total depth in 5 µm steps), were acquired at the end of each imaging session. Z-drift was calculated from the local z-stack and experiments with z-drift of more than 10 µm during the experiment were excluded. The imaging depth of the field of view was confirmed from the full-depth cortical z-stack.

Calcium fluorescence movies were motion corrected for rigid translational errors using an algorithm based on phase correlation. ROI masks of neuronal somata were segmented from motion-corrected movies by (1) creating initial binarized masks using an adaptive fluorescence threshold, (2) applying a succession of morphological operations to fill closed holes and concave shapes, (3) computing a feature vector of each mask that included morphological attributes such as location, area, perimeter, and compactness, (4) combining or eliminating ROIs based on heuristic decisions, including attributes from the feature vectors, and (5) applying a final discrimination step using a binary relevance classifier fed by experimental metadata (e.g. Cre line and imaging depth) as well as the morphological feature vectors. Fluorescence traces were then extracted for each final ROI, which were then neuropil subtracted and corrected for overlapping ROIs by demixing traces. Neuropil contamination into the ROI contributed by the surrounding neuropil was estimated by modeling the measured ROI fluorescence as the sum of the true ROI fluorescence and a weighting of the surrounding neuropil fluorescence, $F_M = F_C + rF_N$, where $F_M$ is the measured fluorescence trace, $F_C$ is the unknown true ROI fluorescence trace that we are trying to estimate, $F_N$ is the fluorescence of the surrounding neuropil and $r$ is the contamination ratio. The contamination ratio was estimated for each ROI by selecting the value for $r$ that minimizes the cross-validation error, $E = \sum_t |F_C - F_M + rF_N|^2$, over four folds. Overlapping ROIs were demixed by modeling the measured fluorescence $F_{it}$ of each pixel $i$ at time $t$ as $F_{it} = \sum_k W_{kit}T_{kt}$, where $W_{kit}$ are time-dependent weighted masks that describe how much of each neuron $k$'s fluorescence is contained in each pixel at each timestep, and $T_{kt}$ is the fluorescence trace of the neurons that we seek to estimate. Reconstruction of calcium movies is modeled as $\sum_i A_{ki}F_{it} = \sum_{k,i} A_{ki}W_{kit}T_{kt}$, where $A_{kt}$ are the binary spatial masks obtained in the earlier segmentation step in which $A_{ki}$ equals 1 if pixel $i$ is in ROI $k$ and equals 0 otherwise. To solve for $T_{kt}$ at each time $t$, we first estimated the weighted masks $W_{kit}$ by projection of the recorded fluorescence $F_{it}$ onto the binary masks $A_{ki}$, then computed the linear least-squares solution $T_{kt}$ to extract each ROI trace's value. To calculate ΔF/F traces from each fluorescence trace, a fluorescence baseline was determined by median filtering the fluorescence trace with a window of 180 s (5401 samples); the ΔF/F trace was then produced by subtracting the fluorescence baseline from the original trace followed by dividing the fluorescence baseline. To prevent very small or negative baselines, we set the baseline as the maximum of the median filter-estimated baseline and the standard deviation of the estimated noise of the fluorescence trace. All analyses of cell responses were performed on L0 penalized detected events (*Jewell and Witten, 2018*; *Jewell et al., 2019*).

Two-photon imaging data was collected from the retinotopic center of primary visual cortex that was identified through mapping during widefield intrinsic signal imaging. *Cux2*-CreERT2;*Camk2a*-tTA;Ai93 and *Vip*-IRES-Cre;Ai148 were imaged at 175 um below the cortical surface in layer 2/3; *Sst*-IRES-Cre;Ai148 mice and *Rorb*-IRES2-Cre;*Camk2a*-tTA;Ai93 mice were imaged at 275 um below the cortical surface in layer 4; *Rbp4*-Cre_KL100;*Camk2a*-tTA;Ai93 mice were imaged at 375 um below

the cortical surface in layer 5; and *Ntsr1*-Cre_GN220;Ai148 mice were imaged at 550 um below the cortical surface in layer 6. (These Cre lines and imaging depths match those used in the Allen Brain Observatory.) Some mice were imaged in two different fields of view at the same depth; the sample sizes for number of imaging sessions and mice are given in *Figure 1—source data 1*. Some mice were imaged in multiple sessions; in cases in which a subset of cells was imaged in multiple sessions, only data from the first imaging session for each cell was analyzed. Mice were excluded for evidence of epileptiform activity, and individual imaging sessions were failed if there were signs of bleaching, saturation, excessive z-drift, or animal stress, among other factors.

## Visual stimulus

As experimental sessions took place on the same data collection pipeline as the Allen Brain Observatory, visual stimulus monitor calibration and positioning (ASUS PA248Q LCD monitor with 1920×1200 pixels; center of monitor was 118.6 mm lateral, 86.2 mm anterior, and 31.6 mm dorsal to the right eye; normal distance from the right eye to center of monitor was 15 cm) were identical. Each monitor was gamma corrected and had a mean luminance of 50 cd m$^{-2}$. Spherical warping was applied to all stimuli to ensure constant spatial and temporal frequencies across the monitor as seen from the mouse's perspective. See *de Vries et al., 2020* for further visual stimulus presentation details. The stimulus consisted of a full field drifting sinusoidal grating that was presented at a single spatial frequency (0.04 cycles/degree) and temporal frequency (1 Hz), eight directions uniformly distributed in 45 degree increments (0 degrees = horizontal front-to-back motion), and six contrasts (5%, 10%, 20%, 40%, 60%, and 80%). Direction of motion was always orthogonal to the orientation of the grating. Each grating was presented for 2 s, followed by 1 s of mean luminance gray before the next grating. Each grating condition (direction, contrast combination) was presented 15–24 times. Trials were randomized with 30 randomly interleaved blank (i.e. mean luminance gray, zero contrast) trials.

## Analysis

### Statistical test for responsiveness

A chi-square test for independence was used to determine significantly responsive cells to the drifting grating stimulus set. A chi-square test statistic was computed $\chi^2 = \sum_{i=0}^{n} \frac{(E_i - O_i)^2}{E_i}$, where $O_i = \frac{1}{m_i} \sum_{j=0}^{m_i} R_{i,j}$ is the observed average response ($R$) of the neuron over $m$ presentations of a grating stimulus of a particular condition (i.e. direction-by-contrast pair or blank, n = 49 total conditions), and $E_i = \frac{\sum_{i}^{n} \sum_{j}^{m_i} R_{i,j}}{\sum_{i}^{n} m_i}$ is the expected (grand average) response per stimulus presentation. A p-value was then calculated for each cell by comparing the test statistic against a null distribution of 200,000 test statistics, each computed from the cell's responses after shuffling (with replacement) cell responses across all presentations.

### Response significance by stimulus condition and test for suppression by contrast

The distribution of responses to stimulus presentations varied substantially across cells. A statistical measure was used to normalize response magnitudes. The mean blank-subtracted response to a given stimulus condition was calculated as: $\bar{R} = \frac{1}{m_i} \sum_{j=0}^{m_i} R_{i,j} - \frac{1}{m_{blank}} \sum_{j=0}^{m_{blank}} R_{blank,j}$. Then, a bootstrapped null distribution of such mean (blank-subtracted) condition responses was generated by sampling with replacement from all of the cell's responses across all stimulus presentations. The percentiles of each cell's observed mean condition response within its own bootstrapped distribution was then computed. Cells were determined to be suppressed by high contrast if this percentile for the peak direction grating condition at 80% contrast was below 0.05.

## Orientation and direction selectivity metrics

Global orientation selectivity was computed from mean extracted event responses to drifting gratings, at the cell's preferred contrast as,

$$gOSI = \frac{\sum R_\theta e^{i\theta}}{\sum R_\theta}$$

where $\theta$ is the direction of grating movement, and $R_\theta$ is the mean response to that direction of motion.

Direction selectivity was computed from mean extracted event responses to drifting gratings, at the cell's preferred contrast, as

$$DSI = \frac{R_{pref} - R_{null}}{R_{pref} + R_{null}}$$

where $R_{pref}$ is a cell's mean response in its preferred direction (i.e. largest response-evoking direction) and $R_{null}$ is its mean response to the opposite direction.

## Contrast preference metric

Contrast preference was computed from mean extracted event responses to drifting gratings, at the cell's preferred direction, as

$$c_{CoM} = e^{\left(\frac{\sum R_c \ln c}{\sum R_c}\right)}$$

where c is the contrast of the drifting grating, $R_c$ is a cell's mean response at contrast c, and $c_{CoM}$ is the log-scaled center of mass of the cell's contrast response tuning.

## Bias in population direction preference

The direction and magnitude of bias in direction preference for a population of cells (e.g. all cells recorded from one mouse or all cells recorded from all mice of a particular Cre line) was calculated as the direction and magnitude of the vector sum of the direction preferences of the cells that comprise the population, at a particular contrast as,

$$\theta_{bias} = \tan^{-1}\left(\frac{\sum \sin \theta_i}{\sum \cos \theta_i}\right)$$

$$r_{bias} = \frac{1}{n_{cells}}\sqrt{\left(\sum \cos \theta_i\right)^2 + \left(\sum \sin \theta_i\right)^2}$$

where $\theta_i$ is the preferred direction of cell $i$, $n_{cells}$ is the number of cells in the population, $\theta_{bias}$ is the direction of the vector sum over the population, and $r_{bias}$ is the magnitude of the vector sum over the population.

## Stimulus tuning conditioned on locomotion behavior

As part of the standardized pipeline for the Allen Brain Observatory, mice were held on a running wheel during experimental sessions and locomotion behavior was recorded (See *de Vries et al., 2020* for further run speed measurement details). The mean running speed was calculated for each trial over the same time window as the mean cellular response was calculated. Trials for which the mean running speed was greater than or equal to 1 cm/s were categorized as running trials, whereas trials for which the mean running speed was below 1 cm/s were categorized as stationary trials. The mean and standard error of the mean event magnitude for each contrast and direction condition shown in *Figure 3* was calculated separately for running and stationary trials. The criterion for a cell to be included in the calculation for a given direction-by-contrast condition was that the mouse had to be running for a minimum of four trials *and* be stationary for a minimum of four trials of that condition. At least three responsive neurons needed to be present to include an experiment in this analysis.

## Contrast response function fitting and model comparison

Event responses as a function of contrast, at a cell's preferred direction, were fit to a rising sigmoid ('high pass'), a falling sigmoid ('low pass'), and the product of one rising and one falling sigmoid ('band pass').

$$R_{high\,pass}\left(c;h,b,s,c_{50}^r\right) = b+h\frac{1}{1+e^{-s\left(c-c_{50}^r\right)}}$$

$$R_{low\,pass}\left(c;h,b,s,c_{50}^f\right) = b+h\frac{1}{1+e^{s\left(c-c_{50}^f\right)}}$$

$$R_{band\,pass}\left(c;h,b,s,c_{50}^r,c_{50}^f\right) = b+h\left(\frac{1}{1+e^{-s\left(c-c_{50}^r\right)}}\right)\left(\frac{1}{1+e^{s\left(c-c_{50}^f\right)}}\right)$$

where $c$ is the contrast, $c_{50}^r$ is the contrast at which the response rises halfway between the base and height, $c_{50}^f$ is the contrast at which the response falls halfway between the base and height, $b$ is the lowest response, $h$ is the response amplitude, and $s$ is the slope of the sigmoid (fixed at $s = 10$). The best fit model was determined by calculating the Akaike Information Criterion (AIC) for each model and selecting the model with the lowest AIC.

The AIC can be calculated as:

$$AIC = 2k - 2\ln\mathfrak{L}$$
$$\mathfrak{L} = \prod_{contrasts}\prod_{trials}\mathfrak{R}\left(R_c^i|\mu=\hat{R}_c,\sigma_R^2\right)$$
$$\ln\mathfrak{L} = -\frac{1}{2\sigma_R^2}\sum_{contrasts}\sum_{trials}\left(R_c^i-\hat{R}_c\right)^2+constant$$

where $k$ is the number of parameters fit in the model, $\mathfrak{L}$ is the likelihood of observing the responses given the fitted model and response distribution, $R_c^i$ is the cell's response to a grating stimulus of contrast c (at the cell's preferred direction) on trial $i$, $R_c$ is the response predicted by the model to a grating stimulus of contrast c, $\sigma_R^2$ is the variance of all of the cell's responses, and $\mathfrak{R}$ is the normal distribution. In practice, it is more convenient to directly calculate the log-likelihood than to calculate the likelihood and subsequently take the log, and the constant can be ignored for model selection since the same constant applies to all models being compared.

Due to the non-normal response distribution, possibly arising from calcium imaging as well as an underlying non-normal spiking distribution, we bootstrapped the log-likelihood rather than assume normality. Therefore, the likelihood was calculated numerically by shuffling responses across trials 1000 times and calculating the sum of square residuals from the predicted responses as $SS = \sum_{contrasts}\sum_{trials}\left(R_c^i-R_c\right)^2$ for each shuffle. The likelihood was taken as the fraction of shuffles for which $SS$ was greater than the observed $SS$.

## Generalized linear model

We constructed Generalized Linear Models, specifically a Poisson (i.e. exponential function) models, to predict the population response of each neuron type (e.g. VIP neurons) on each trial from stimulus contrast, stimulus direction, locomotion state (i.e. binary run or not run variable), and the interactions between these terms. The model was

$$\hat{R}_{b,r,d,c} = e^{\left(w_b a_b+w_r a_r+\sum_d a_d\left(w_d+w_{d,r}a_r\right)+\sum_c a_c\left(w_c+w_{c,r}a_r\right)+\sum_d\sum_c a_d a_c\left(w_{d,c}+w_{d,c,r}a_r\right)+k\right)}$$

where $\hat{R}_{b,r,d,c}$ is the predicted response for a trial, $w$ terms are the weights of the model, $a$ terms are binary variables that equal 1 if the trial attribute is true and equal 0 otherwise; the trial attributes are blank ($b$), run state ($r$), stimulus direction ($d$), and stimulus contrast ($c$); and $k$ is a constant. The weights of the model were computed by minimizing the cost function $L$ using iterative reweighted least squares,

$$L = SSE + \lambda l_1$$

where SSE is the reconstruction error.

$$SSE = \sum_{b,r,d,c} \left( R_{b,r,d,c} - \hat{R}_{b,r,d,c} \right)^2$$

and $l_1$ is an L1-regularization penalty that serves to identify only weights that significantly contribute to neuronal responses,

$$l_1 = |w_b| + |w_r| + \sum_d \left( |w_d| + |w_{d,r}| \right) + \sum_c \left( |w_c| + |w_{c,r}| \right) + \sum_d \sum_c \left( |w_{d,c}| + |w_{d,c,r}| \right)$$

The strength of regularization, $\lambda$, was determined through leave-one-out cross validation in which one experimental session was left out for each fold.

## Stabilized supralinear network (SSN) model

The SSN was modeled as a ring network, largely maintaining the basic architecture and dynamics described in *Rubin et al., 2015* but deviating primarily in the diversity of inhibitory neurons and distributions of connections between neuron populations (including untuned inhibitory connections, described below). Our network consisted of one excitatory population (representing layer 2/3 CUX2 pyramidal neurons) and three inhibitory populations (representing PV, SST, and VIP interneurons, respectively). The ring network structure was imposed by providing each excitatory neuron with external ('sensory') excitatory input that had Gaussian tuning with the mean (i.e. peak/preferred direction) corresponding to the neuron's position on the ring and standard deviation of 30 degrees; PV neurons also received external input which was not tuned (i.e. all PV cells receive input of equal strength). The entire network covered 180 degrees of orientation (or direction). The strength of external input was intended to represent a monotonically-increasing function of stimulus contrast, though no specific relationship between input magnitude and contrast is claimed here.

Connections between neurons also had Gaussian tuning that depended on the difference between the orientation preferences of the pre- and post-synaptic neurons (*Figure 5b*). The distributions of recurrent excitatory connections onto CUX2 cells and excitatory connections onto Vip cells were narrow (standard deviation of 30 degrees) compared to the distributions of connections to and from PV and Sst cells (standard deviation of 100 degrees).

The network consisted of 184 excitatory neurons, 40 PV neurons, 15 SST neurons, and 15 VIP neurons. The excitatory population had 180 neurons with uniform 1-degree spacing of peak directions to tile the ring, plus four extra neurons with peak direction of zero degrees to capture the slight bias of the CUX2 neurons. All model VIP neurons had a peak direction of zero degrees to capture the strong bias for front-to-back motion observed for VIP neurons. In addition, all SST and PV model neurons also had a peak direction of zero degrees, though the very broadly-tuned inputs to these neurons results in a much weaker bias of net input to these neurons than the bias to VIP neurons. All neurons were implemented as rate models with firing rate that was a rectified quadratic function of the summed input to the neuron,

$$r_{ss}(I) = \begin{cases} kI^2 & I > 0 \\ 0 & I \leq 0 \end{cases}$$

where $I$ is the input strength, $r_{ss}$ is the steady state firing rate, and $k$ is a constant of proportionality. For ease of comparison with the SSN models developed by *Rubin et al., 2015*, we used $k = 0.04$ for all models.

For a given external input, the firing rates of all neurons in the network were obtained by evolving the network in time, with dynamics:

$$\begin{aligned} \dot{r} &= r_{ss}(I_{sum}(t)) - r(t) \\ I^j_{sum}(t) &= I^j_{sp} + \sum_i W_{i,j} r^i(t) \end{aligned}$$

where $r(t)$ is the time-dependent firing rate, $\dot{r}$ is the time derivate of the neuron's firing rate, $r_{ss}$ is the steady state firing rate that varies in time based on the inputs to the neuron, $I^j_{sum}$ is the net input to neuron $j$, $I^j_{sp}$ is a constant spontaneous input to neuron $j$, and $W_{i,j}$ is the connection strength from presynaptic neuron $i$ onto postsynaptic neuron $j$. To provide a spontaneous activity to the network,

and account for the higher spontaneous activity of VIP neurons (*Roux and Buzsáki, 2015*), we set $I_{sp}^{CUX2} = I_{sp}^{PV} = I_{sp}^{SST} = 2$ and $I_{sp}^{VIP} = 10$. The network is evolved with Euler integration with updates of $r^j = \frac{t}{\tau^j}\dot{r}^j$ at each time step of $t = 0.1\ ms$, where the time constants of the different neuron types are $\tau^{CUX2} = \tau^{SST} = \tau^{VIP} = 20\ ms$ and $\tau^{PV} = 10\ ms$.

We calculated the stability of the steady state of activity at zero degrees with respect to a spatially homogenous perturbation of the inputs. The stability matrix is, in the spatial Fourier domain,

$$J(x,n) = g_i(x)W_{ij}G_{ij}(x,n)$$

where $x$ is the postsynaptic location in degrees, $n$ is the spatial frequency corresponding to the orientation difference between the presynaptic and postsynaptic cells, $g_i(x)$ is the postsynaptic gain at its steady-state rate, $W$ is the weight matrix and $G_{ij}(x,n)$ is the Fourier transform of the wrapped Gaussian connectivity profile:

$$G(x,n) = 30\sqrt{\frac{2}{\pi}}\begin{pmatrix} e^{-2(n\pi\sigma_e)2} & e^{-2(n\pi\sigma_b)2} & e^{-2(n\pi\sigma_b)2} & e^{-2(n\pi\sigma_e)2} \\ e^{-2(n\pi\sigma_b)2} & e^{-2(n\pi\sigma_b)2} & e^{-2(n\pi\sigma_b)2} & e^{-2(n\pi\sigma_e)2} \\ e^{-2(n\pi\sigma_b)2} & e^{-2(n\pi\sigma_b)2} & e^{-2(n\pi\sigma_b)2} & e^{-2(n\pi\sigma_e)2} \\ e^{-2(n\pi\sigma_e)2} & e^{-2(n\pi\sigma_b)2} & e^{-2(n\pi\sigma_b)2} & e^{-2(n\pi\sigma_e)2} \end{pmatrix}$$

where $\sigma_e = 30$ degrees is the projection width for E→E, E→VIP and VIP projections and $\sigma_b = 100$ degrees is the projection width for the remaining inhibitory projections. If $EE_{stability} = J_{00}(x,n) - 1$ is greater than zero, the network at orientation $x$ is in an inhibitory-stabilized state with respect to perturbations at spatial frequency $n$.

## Acknowledgements

The authors wish to thank the Allen Institute founder, Paul G Allen, for his vision, encouragement, and support.

## Additional information

### Funding

| Funder | Author |
|---|---|
| Allen Institute for Brain Science | Daniel J Millman |
| | Gabriel Koch Ocker |
| | Shiella Caldejon |
| | India Kato |
| | Josh D Larkin |
| | Eric Kenji Lee |
| | Jennifer Luviano |
| | Chelsea Nayan |
| | Thuyanh V Nguyen |
| | Kat North |
| | Sam Seid |
| | Cassandra White |
| | Jerome Lecoq |
| | Clay Reid |
| | Michael A Buice |
| | Saskia EJ de Vries |

The funders had no role in study design, data collection and interpretation, or the decision to submit the work for publication.

### Author contributions

Daniel J Millman, Formal analysis, Methodology, Writing - original draft, Writing - review and editing; Gabriel Koch Ocker, Formal analysis, Methodology, Writing - review and editing; Shiella Caldejon, India Kato, Josh D Larkin, Eric Kenji Lee, Jennifer Luviano, Chelsea Nayan, Thuyanh V Nguyen, Kat North, Sam Seid, Cassandra White, Investigation; Jerome Lecoq, Supervision, Methodology; Clay Reid, Supervision; Michael A Buice, Conceptualization, Supervision, Writing - review and

editing; Saskia EJ de Vries, Conceptualization, Formal analysis, Supervision, Methodology, Writing - review and editing

### Author ORCIDs
Daniel J Millman (iD) https://orcid.org/0000-0002-6255-6085
Gabriel Koch Ocker (iD) http://orcid.org/0000-0001-9627-9576
Eric Kenji Lee (iD) http://orcid.org/0000-0002-7166-0909
Clay Reid (iD) http://orcid.org/0000-0002-8697-6797
Saskia EJ de Vries (iD) https://orcid.org/0000-0002-3704-3499

### Ethics
Animal experimentation: All animal procedures were approved by the Institutional Animal Care and Use Committee (IACUC) at the Allen Institute for Brain Science.

### Decision letter and Author response
Decision letter https://doi.org/10.7554/eLife.55130.sa1
Author response https://doi.org/10.7554/eLife.55130.sa2

## Additional files

### Supplementary files
• Transparent reporting form

### Data availability
The data generated and analyzed in this study are available on DANDI: Distributed Archives for Neurophysiology Data Integration. All analyses were performed using custom scripts written in Python 2.7, using NumPy, SciPy, Pandas, Matplotlib, statsmodel, and Scikit-learn. Analysis code is available at https://github.com/AllenInstitute/Contrast_Analysis (copy archived at https://archive.softwareheritage.org/swh:1:rev:c7ddda11647093e8a0173dbd2a1986ac6239c821/). Event extraction was performed using FastLZeroSpikeInference available at https://github.com/jewellsean/FastLZeroSpikeInference.

The following dataset was generated:

| Author(s) | Year | Dataset title | Dataset URL | Database and Identifier |
|---|---|---|---|---|
| Millman DJ, de Vries SE | 2020 | Allen Institute – Contrast tuning in mouse visual cortex with calcium imaging | https://dandiarchive.org/dandiset/000039/draft | DANDI, 000039 |

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
