## [Decision Letter]

**Acceptance summary:**

Millman et al. show new insights into the function and visual response properties of a main subclass of vasoactive intestinal peptide-expressing (VIP) interneurons in area V1 of the mouse. They find that VIP interneurons are most strongly activated by low-contrast stimuli with front-to-back motion, which is congruent with self-motion that occurs during locomotion, another main factor that is known to drive VIP interneuron activity. The authors develop a model suggesting that this accounts for the suppression of SSt+ (Somatostatin) interneurons at low luminance-contrasts, and the surprising preference of Layer 2/3 pyramidal cells for low-contrast stimuli.

**Decision letter after peer review:**

Thank you for submitting your article "VIP interneurons selectively enhance weak but behaviorally-relevant stimuli" for consideration by *eLife*. Your article has been reviewed by four peer reviewers, one of whom, Martin Vinck, is a member of our Board of Reviewing Editors, and the evaluation has been overseen by Kate Wassum as the Senior Editor. The other reviewers have opted to remain anonymous.

The reviewers have discussed the reviews with one another and the Reviewing Editor has drafted this decision to help you prepare a revised submission.

Summary:

VIP interneurons are a class of GABAergic interneurons that can disinhibit pyramidal neurons via inhibition of SST interneurons. Millman et al. describe a novel role for VIP-driven disinhibition of pyramidal cells in the mouse primary visual cortex (V1). They report surprising and novel cell type-specific responses in the mouse visual cortex. First, VIP-positive interneurons show stereotyped direction tuning, strikingly different from the pyramidal cell population, suggesting that inputs to these neurons are related to a top-down input that signals behavioral relevance of the stimulus. The finding that VIP responses are enhanced by locomotion fits with this hypothesis. Second, VIP responses are suppressed at high contrast. Finally, a model based on the SSN architecture can reproduce some of the major features of the data. Together, this work supports the notion that VIP neuron activity increases sensitivity to weak inputs, whereas SST neuron activity decreases sensitivity to strong inputs and maintains network stability. This interesting conjunction leads to heightened sensitivity to detect low-contrast objects or obstacles approaching head-on during locomotion. Thus, VIP neurons promote high gain in the local microcircuit at low contrast conditions.

General assessment:

All reviewers agreed that the study has substantial merits and makes an important, novel contribution to the field.

Essential revisions:

The reviewers raise a number of concerns that must be adequately addressed before the paper can be accepted. Some of the required revisions will require further data analysis.

1) Data representation and interpretation of calcium signals

The data shown are highly overprocessed. The authors should include raw data traces and images, and describe their methods comprehensively in the paper. Data in Figure 1 should be shown with proper normalization (percent change) as well as raw data, and not as percentiles. While the authors use an established "pipeline" for data collection, they should include raw calcium imaging traces and images (at least in the Materials and methods), They should describe their methods comprehensively in the paper. The paper should be self-contained, and the authors should provide a representation of how fluorescence images were converted into the color maps in Figure 1. Inclusion of raw data is necessary for readers to get a sense of what is measured and possible caveats. For example, it could be possible that VIP neurons have saturating response at high contrast and the calcium does not decrease to baseline between trials, resulting in what looks like a suppressed response. A further consideration is the fact that imaging depths vary throughout the manuscript, and GABAergic neuronal activity can be substantially higher than pyramidal cell activity, which should have a "smoothing" impact on calcium traces. The authors should better explain how they took this into consideration in their estimation of various metrics. The authors should further motivate why data was averaged over running and stationary trials in Figure 1.

2) Authors should discuss how GCaMP signals relate to spikes across different cell types (for example see Khan et al., Nat Nsci 2018, Figure 2E and Supplementary Figure 6, where a comparison is made with calcium imaging and extracellular recording in Vip, PV and Sst neurons) (and Huang et al-Biorxiv).

3) It is unclear why the authors used Cre lines to image different pyramidal neuron populations, along with imaging at different depths. A possible reason could be to avoid contamination from dendrites of neurons in different layers. If this is the reason it should be stated in the paper. In addition, they asked for something to be said about the idiosyncrasies of the cre lines in light of recent scRNAseq data.

4) Response percentiles in Figure 1B are reported, but these are never corrected for the many multiple comparisons, such that the significance is merely a normalized measure of strength. However percentiles are deceiving because they hide the amplitude of the responses. Data should be represented as percent change to obtain an idea about the effect sizes shown in this study, because a small effect might correspond to a large normalized measure of strength. In general, the color map in Figures 1 and 2 are unconventional and needs to be justified better. It is hard to judge how strong the responses are overall, and what the differences between cell types mean.

With respect to questions about effect size: reviewers asked whether the small numbers throughout Figure 1B are significant, and pointed out that some of these cells appear to be barely responsive. Reviewers were surprised about the seemingly tiny responses in Figure 2O and P, and questioned whether this can be a meaningful code for direction? What do these magnitudes mean in terms of spike numbers?

5) Averaging across the raw magnitudes in Figure 2 across neurons could be a highly insensitive analysis and lead to conclusions that are dominated by a few neurons with high baseline firing rates. A standard way to analyze this kind of data would be to normalize the responses per neurons by for example dividing by the number of baseline events. Further why are the response magnitudes in Figure 2 not normalized as in Figure 1B?

6) Figure 1C: It appears that no multiple-comparison corrected statistics are provided on the differences in population orientation tuning between different lines. Given that this study is explorative, inferential statistics should be performed to compare mice lines and should be corrected for multiple comparisons (contrasts x cre lines). The text sections, "CUX2 neurons in layer 2/3 showed direction bias toward front-to-back motion at 5% and 10% contrast but not at higher contrasts (Figure 1C);. pyramidal neurons in deeper layers did not 83 have direction bias." seems not motivated by statistics. A statistical test should be provided for the difference between low and high contrasts. Furthermore, the statistical difference between superficial layer and deep layer is suggested by the text, but appears to be never explicitly tested. The authors should carefully check that all of their claims on differences between groups or conditions are supported by multiple-comparison corrected statistics. A general problem in this study is that claims on differences between lines are made, but never made explicit through testing. An example is the description: "79: Contrast preference among pyramidal neurons systematically varied across cortical layers, exhibiting a progression from a mixture of low and high contrast-preferring neurons in layer 2/3 to almost exclusively high contrast-preferring neurons in layers 5 and 6 (Figure 1F, G)." The claim of differences among cell lines should be substantiated with statistics that test whether the different mouse lines are significantly different (the unit of analysis should be the mouse).

7) Claims on interactions between variables

Claims about the interactions between variables needs to be demonstrated by statistics. In principal cells, direction tuning is canonically thought to be largely invariant to contrast tuning. The authors should explicitly analyse whether the direction tuning for VIP interneurons is invariant to contrast, and whether the front-to-back preference is indeed specific for low contrast (as the authors suggest) – i.e. test against the alternative that contrast simply gain-modulates the direction tuning. A similar question pertains to the interaction between locomotion and direction tuning. The manuscript description of Figure 2 suggests an interaction between the Locomotion/Stationary variable with contrast and direction tuning. However this does not appear to be statistically quantified, and that it is possible that contrast tuning curves are modulated in a multiplicative way (i.e. invariant to locomotion). A similar comment pertains to, "This analysis demonstrates a substantial enhancement of responses to low contrast visual stimuli during locomotion that is specific to layer 2/3 pyramidal neurons and VIP neurons." There could be a general multiplicative increase in firing across cell lines and contrasts, and Figure 2 does not demonstrate that low contrast visual stimuli are enhanced specifically during locomotion.

8) Imaging depths and layers

The authors need to carefully check their depths and laminar assignments and update the text on this. The authors should more carefully discuss in the manuscript what type of SSt and VIP cells were imaged and discuss the implications of interneuron heterogeneity. Is there any particular reason the mice from SSt-Cre mouse line were imaged in layer 4 as stated in the Materials and methods? As the authors know, dendrite-targeting Martinotti cells are more likely to be found in Layers 2/3 and 5 (Munoz et al., 2017, Science), and the Agmon Lab found that SSt-expressing neurons from layer 4 barrel cortex had unique and different electrophysiological properties from other SSt neurons (Ma et al., 2006, J.Neurosci). If indeed, this is the only imaging plane used for in the SSt-Cre mouse line, how come the authors describe the activity of SSt neurons in layers 2/3 (main text, fourth paragraph)? Similarly for layer 4 VIP cells activity (same lines). Generally there is a bit of confusion with the reported imaging depths and the authors should comment on this. In particular the laminar assignment of the SSt neurons to be clarified. According to the depth 275 micrometer one would say these are layer 2/3, but the authors write layer 4. In general, the laminar assignment based on depth should be justified if it is not based on a layer-specific mouse line.

9) Generalizability

Due to the focus on very specific stimuli, the scope of the study is limited and the results do maybe not generalize beyond large drifting gratings. For instance, it is unclear whether these findings will hold up for smaller stimuli. Other stimulus parameters are not explored systematically (spatial frequency or temporal frequency). Stimulus size is ignored, even though stimulus size could have dramatic effects on the findings presented here. The authors need to discuss these limitations in the manuscript. In particular they should comment on the issue of size tuning, which is missing in the manuscript. Previous studies have shown a very clear size dependence of VIP and SSt neurons (Adesnik and Scanziani, 2012; DiPoppa et al., 2018). The contrast dependence of the VIP and SSt neurons is likely strongly dependent on stimulus size. For instance VIP neurons have small receptive fields (DiPoppa et al., 2018). Because surround modulation is contrast dependent, suppression of PC L2/3 cell activity and VIP activity likely depends on the stimulus size. The authors to discuss these limitations.

10) Interpretation of SSt direction tuning and VIP suppression

The authors should discuss the interpretation of some of the main effects. In particular, reviewers wondered why VIP neurons are suppressed at high contrast? They also wondered why the response of SST neurons at low contrast is not direction modulated – given that the VIP neurons have strong direction tuning? Do VIP interneurons indeed suppress the SSt neurons recorded here? This assumption is being made but it is never tested or argued for these specific laminar recordings from the literature.

11) Optic Flow

The interpretation of optic flow appears to be problematic and needs to be discussed. The authors interpret their data in terms of optic flow related to locomotion. However, the peak of VIP neurons occurs at 45 degrees rather than 0 degrees, but the authors write that 0 degrees corresponds to front-to-back motion (main text, second paragraph). Does the interpretation of the authors make sense given this discrepancy?

12) Modelling:

There are major concerns about parameter and model selection. Several findings hard to account for with the model. The description of the model needs to be substantially improved. The tuning of VIP cells in 3b is much wider than that observed in the data (1b), and this could be a major problem for the model. The authors should explain this. The authors should make a better effort to explain the model better in the text. For the modelling, authors need to provide more explanation on why SSNs were selected, and how they work etc. In the current version of the paper, it is necessary to refer to the very difficult Rubin et al., 2015 paper to understand what the model does. Also, reviewers would like to see that the canonical results from Rubin et al. still work in the revised model, and commented that this is an essential addition to the paper. The authors should include a discussion of the process of parameter selection for the model (how much fine tuning was required, what happens when connection weights deviate somewhat etc). In general, the scope of the model to be somewhat limited given that model parameters are not systematically explored or fitted based on the data. Further, the model makes strong assumptions on the inputs that these different neuron types presumably receive from external drive. These limitations need to be discussed. The main text to be improved with a more in-depth explanation for using SSNs (stabilized supra-linear network). Finally, rectified quadratic response functions for neurons are not common in RNNs, and should be motivated. Also, no explanation/citation is provided for the choice of parameters (input, connections) for PV neurons.

With respect to the conclusion – "These results indicate that Vip-mediated disinhibition is capable of producing substantial increases in gain at low contrast despite low activity of the intermediate SST neuron population." The authors should explain why this is not a circular argument and a strange way to summarize the results of the model. Reviewers pointed out that it is clear from Figure 3D that the suppression of Sst activity by Vip neurons at low contrast is what enables supra-linear responses of Cux2 neurons at low contrast.

The bias of direction tuning in L2/3 neurons is very weak. How can the authors account for the contrast enhancement of these neurons in their VIP model? Should this enhancement not be extremely specific to the 45 degrees angle?

---

## [Author Response]

Essential revisions:The reviewers raise a number of concerns that must be adequately addressed before the paper can be accepted. Some of the required revisions will require further data analysis.1) Data representation and interpretation of calcium signalsThe data shown are highly overprocessed. The authors should include raw data traces and images, and describe their methods comprehensively in the paper. Data in Figure 1 should be shown with proper normalization (percent change) as well as raw data, and not as percentiles. While the authors use an established "pipeline" for data collection, they should include raw calcium imaging traces and images (at least in the Materials and methods), They should describe their methods comprehensively in the paper. The paper should be self-contained, and the authors should provide a representation of how fluorescence images were converted into the color maps in Figure 1. Inclusion of raw data is necessary for readers to get a sense of what is measured and possible caveats. For example, it could be possible that VIP neurons have saturating response at high contrast and the calcium does not decrease to baseline between trials, resulting in what looks like a suppressed response. A further consideration is the fact that imaging depths vary throughout the manuscript, and GABAergic neuronal activity can be substantially higher than pyramidal cell activity, which should have a "smoothing" impact on calcium traces. The authors should better explain how they took this into consideration in their estimation of various metrics.

We have made several, substantial changes to the main text, Materials and methods section, and figures to address these points. Together, we believe that these additions will resolve these issues.

– The revised manuscript includes a new Figure 1 that shows fluorescence traces for four example neurons, of the key Cre lines, as well as stepwise transformations to “events” in the fluorescence traces and, finally, stimulus response magnitudes and tuning curves.

– We changed the normalization used in Figure 2B (previously Figure 1B), which we discuss further in our response to point 4 below.

– We have expanded the description of the processing of calcium imaging data in the Materials and methods section to make this paper self-contained.

– We have added additional detail on the imaging depths in the Materials and methods section and have carefully documented the imaged layer in the text and figures. See our reply to point 8 below for an extended discussion of imaging depths.

– We show that the fluorescence traces of VIP neurons are substantially suppressed in response to high contrast gratings, not saturated, in Figure 2—figure supplement 2. Therefore, the suppression of VIP responses as measured in events does not result from fluorescence saturation during high contrast grating presentations.

– As can be seen in the example dF/F traces in Figure 1, we observe clear calcium transients for inhibitory interneurons that are not substantially “smoother” than those of excitatory neurons. We are aware that the high spike rates and calcium buffering in PV+ interneurons can lead to highly smoothed traces, but that does not appear to be the case for the VIP and SST interneurons imaged in this study.

Based on this, we do not believe that our conclusions based on the tuning metrics we calculate (e.g. DSI, gOSI, etc.) are likely to be impacted by higher spontaneous rates of interneurons.

The authors should further motivate why data was averaged over running and stationary trials in Figure 1.

The motivation for this study was to investigate the influence of stimulus contrast and direction on neuron responses across Cre lines and layers. The findings regarding contrast and direction tuning described in Figure 2 (previously Figure 1) which is already a very dense full-page figure, motivate further examination of locomotion. Therefore, we address the influence of locomotion on contrast and direction tuning in the subsequent two figures. In particular, related to reviewers’ point 7 below, we have added a Generalized Linear Model analysis to specifically address the relationship between locomotion and stimulus direction and contrast.

2) Authors should discuss how GCaMP signals relate to spikes across different cell types (for example see Khan et al., Nat Nsci 2018, Figure 2E and Supplementary Figure 6, where a comparison is made with calcium imaging and extracellular recording in Vip, PV and Sst neurons) (and Huang et al-Biorxiv).

Khan et al., 2018, show that the relationship between DF/F transients and spikes, for combined loose path and 2P recordings in cortical slices, using injected GCaMP6 and imaged at an unknown magnification. These show that VIP and SST neurons exhibit the same relationship between DF/F and spikes, while the slope of that relationship is different for PV neurons.

Huang et al., 2019, shows the relationship between multiple excitatory Cre lines in layer 2/3. The paper focuses mostly on differences between GCaMP6s and GCaMP6f, comparing transgenically expressed GCaMP with data for virally expressed GCaMP6 from Chen et al. The different Cre lines used for GCaMP6f (the reporter used in our study) are Cux2-CreERT2 (which we use) and Emx1-IRES-Cre. Within layer 2/3, these are both considered to be pan-excitatory, and the results in Huang et al. reveal that they show the same relationship as each other between spikes and DF/F amplitude.

Further, as Huang et al., 2019, alludes to, and is further examined in Ledochowitsch et al., 2019, the spatial and temporal imaging resolution is a critical parameter in this relationship. The analysis in Ledochowitsch et al., 2019, approximates the imaging resolutions used in our study, but again only considers two pan-excitatory Cre lines within layer 2/3.

Since the two studies use different imaging paradigms (slice vs. in vivo, unknown magnifications, etc.), we are unable to meaningfully compare the data in Khan et al. with Huang et al. Further, neither dataset speaks to potential differences in the relationship of spikes and DF/F between the VIP and SST inhibitory populations and pyramidal neurons, or between different pyramidal Cre lines at different imaging depths. As these are the questions we believe the reviewers are asking, and are most relevant to this study, it is unclear how we can address this without doing a separate study on this specific issue. While we agree that this is an important question for the field, it is beyond the scope of this paper.

3) It is unclear why the authors used Cre lines to image different pyramidal neuron populations, along with imaging at different depths. A possible reason could be to avoid contamination from dendrites of neurons in different layers. If this is the reason it should be stated in the paper.

The reviewers are precisely correct that the Cre lines were used in order to target neurons in specific layers without contamination from processes of neurons in different layers. We have now stated this point in the second paragraph of the main text.

In addition, they asked for something to be said about the idiosyncrasies of the cre lines in light of recent scRNAseq data.

Extended Figure 8 of Tasic et al., 2018, provides a summary table of the correspondence between Cre lines (including all 6 used here) and transcriptomic neuron subtypes. We have now discussed the transcriptomic types labeled by the Cre lines used in this study in the Materials and methods under the “Experimental Animals” section (second paragraph) and discuss the potential relationships between transcriptomic types and contrast tuning in the sixth paragraph of the main text. Briefly, scRNA-seq data has shown that the Vip-Cre and Sst-Cre lines provide broad coverage of VIP and SST neuron transcriptomic subtypes (16 and 21 subtypes). In layer 2/3, Cux2-CreERT2 labels all three excitatory neuron transcriptomic subtypes. Layer 4 contains only a single transcriptomic neuron type, which is sampled by the Rorb-Cre line. Rbp4-Cre_KL100 labels all twelve layer 5 neuron transcriptomic subtypes. Ntsr1-Cre labels all six layer 6 corticothalamic neuron transcriptomic subtypes.

4) Response percentiles in Figure 1B are reported, but these are never corrected for the many multiple comparisons, such that the significance is merely a normalized measure of strength. However percentiles are deceiving because they hide the amplitude of the responses. Data should be represented as percent change to obtain an idea about the effect sizes shown in this study, because a small effect might correspond to a large normalized measure of strength. In general, the color map in Figures 1 and 2 are unconventional and needs to be justified better. It is hard to judge how strong the responses are overall, and what the differences between cell types mean.

We understand that the metric we had used, the percentile of the response in a bootstrapped distribution, was a source of confusion. The motivation behind this normalization is analogous to z-scoring the responses, with the additional step of generating a baseline distribution through bootstrapping rather than assuming a normal distribution since the responses with calcium imaging are non-Gaussian and highly skewed. Given the confusion, however, we have followed the reviewers’ suggestion and changed the metric to be fractional change.

The color map used is “RdBu” for the Matplotlib python package. We chose this color map specifically because it is perceptually uniform.

With respect to questions about effect size: reviewers asked whether the small numbers throughout Figure 1B are significant, and pointed out that some of these cells appear to be barely responsive. Reviewers were surprised about the seemingly tiny responses in Figure 2-0 and P, and questioned whether this can be a meaningful code for direction? What do these magnitudes mean in terms of spike numbers?

Note that this figure shows population average responses. The panels in question are for pyramidal neurons which have narrow direction tuning (relative to SST neurons) and varied peak directions. Therefore, we expect the response strength *averaged over neurons* to any direction in particular to be small. Also, we suspect that the potentially unfamiliar units we used (event magnitude per frame) led to the interpretation that the responses were very small. We have changed the units to “event magnitude per second” and show it as a percent rather than fraction. The relationship between these response magnitudes and the calcium fluorescence traces is now illustrated for example neurons in a new Figure 1.

5) Averaging across the raw magnitudes in Figure 2 across neurons could be a highly insensitive analysis and lead to conclusions that are dominated by a few neurons with high baseline firing rates. A standard way to analyze this kind of data would be to normalize the responses per neurons by for example dividing by the number of baseline events.

We have added a new supplementary figure (Figure 3—figure supplement 1) in which we show the full distribution of single neuron response magnitudes as well as box plots to show the quartiles of the distributions.

Further why are the response magnitudes in Figure 2 not normalized as in Figure 1B?

The normalization in Figure 2B (previously Figure 1B) has changed in response to point 4.

6) Figure 1C: It appears that no multiple-comparison corrected statistics are provided on the differences in population orientation tuning between different lines. Given that this study is explorative, inferential statistics should be performed to compare mice lines and should be corrected for multiple comparisons (contrasts x cre lines). The text sections, "CUX2 neurons in layer 2/3 showed direction bias toward front-to-back motion at 5% and 10% contrast but not at higher contrasts (Figure 1C);. pyramidal neurons in deeper layers did not 83 have direction bias." seems not motivated by statistics. A statistical test should be provided for the difference between low and high contrasts.

The 95% confidence intervals shown in Figure 2C (previously Figure 1C) are now corrected for multiple comparisons and the figure legend has been updated to reflect this change.

Furthermore, the statistical difference between superficial layer and deep layer is suggested by the text, but appears to be never explicitly tested. The authors should carefully check that all of their claims on differences between groups or conditions are supported by multiple-comparison corrected statistics. A general problem in this study is that claims on differences between lines are made, but never made explicit through testing. An example is the description: "79: Contrast preference among pyramidal neurons systematically varied across cortical layers, exhibiting a progression from a mixture of low and high contrast-preferring neurons in layer 2/3 to almost exclusively high contrast-preferring neurons in layers 5 and 6 (Figure 1F, G)." The claim of differences among cell lines should be substantiated with statistics that test whether the different mouse lines are significantly different (the unit of analysis should be the mouse).

The revised manuscript includes a new Figure 2E that shows the results of new statistical analysis of contrast tuning across Cre lines and layers. In this panel, we show bootstrapped confidence intervals on the fractions of low contrast preferring neurons and high contrast preferring neurons as well as pairwise tests between the fractions of low contrast preferring pyramidal neurons across layers. We use the mouse as the unit of analysis and bootstrapping.

7) Claims on interactions between variablesClaims about the interactions between variables needs to be demonstrated by statistics. In principal cells, direction tuning is canonically thought to be largely invariant to contrast tuning. The authors should explicitly analyse whether the direction tuning for VIP interneurons is invariant to contrast, and whether the front-to-back preference is indeed specific for low contrast (as the authors suggest) – i.e. test against the alternative that contrast simply gain-modulates the direction tuning. A similar question pertains to the interaction between locomotion and direction tuning. The manuscript description of Figure 2 suggests an interaction between the Locomotion/Stationary variable with contrast and direction tuning. However this does not appear to be statistically quantified, and that it is possible that contrast tuning curves are modulated in a multiplicative way (i.e. invariant to locomotion). A similar comment pertains to, "This analysis demonstrates a substantial enhancement of responses to low contrast visual stimuli during locomotion that is specific to layer 2/3 pyramidal neurons and VIP neurons." There could be a general multiplicative increase in firing across cell lines and contrasts, and Figure 2 does not demonstrate that low contrast visual stimuli are enhanced specifically during locomotion.

The revised manuscript includes a new section and Figure 4 in which we model responses of VIP, SST, and L2/3 pyramidal neurons with a Generalized Linear Model to investigate the contribution of stimulus direction, stimulus contrast, running, and the interactions between these terms. We did not find strong interactions between stimulus direction and contrast for any neuron type, but we did find significant interactions between running and stimulus contrast as well as running and stimulus direction. We have edited the text of the manuscript to more precisely describe the interactions among these variables.

8) Imaging depths and layersThe authors need to carefully check their depths and laminar assignments and update the text on this. The authors should more carefully discuss in the manuscript what type of SSt and VIP cells were imaged and discuss the implications of interneuron heterogeneity. Is there any particular reason the mice from SSt-Cre mouse line were imaged in layer 4 as stated in the Materials and methods? As the authors know, dendrite-targeting Martinotti cells are more likely to be found in Layers 2/3 and 5 (Munoz et al., 2017, Science), and the Agmon Lab found that SSt-expressing neurons from layer 4 barrel cortex had unique and different electrophysiological properties from other SSt neurons (Ma et al., 2006, J.Neurosci). If indeed, this is the only imaging plane used for in the SSt-Cre mouse line, how come the authors describe the activity of SSt neurons in layers 2/3 (main text, fourth paragraph)? Similarly for layer 4 VIP cells activity (same lines). Generally there is a bit of confusion with the reported imaging depths and the authors should comment on this. In particular the laminar assignment of the SSt neurons to be clarified. According to the depth 275 micrometer one would say these are layer 2/3, but the authors write layer 4. In general, the laminar assignment based on depth should be justified if it is not based on a layer-specific mouse line.

The cranial window used for 2P imaging put pressure on the brain, and results in a compression of the cortical layers. The cortical thickness is compressed to ~700um, and the compression is not consistent across layers, such that, for example, layer 5 shows more compression and layer 4 shows less compression, on average (see de Vries, Lecoq, Buice et al. Supplementary Figure 12D and E). Based on this compression, as well as the density of neurons labeled by layer 4 specific excitatory Cre lines (e.g. Rorb, Scnn1a), we estimate that layer 4 begins at ~250um, and layer 5 begins at ~375 um, below surface.

The inhibitory neurons were imaged where they are most densely labeled. For VIP, this is shallower than for SST which we find to be densest at 275-375 μm below the surface. Based on our depth estimate, this puts the SST somata in layer 4, while the VIP neurons that are densest at ~175 μm are in layer 2/3. We believe, related to lines 96-98, that ambiguous wording led to the misinterpretation that VIP and SST neurons were imaged in both layers 2/3 and 4 – this wording has now been changed to clarify that *pyramidal neurons* were imaged in layers 2/3 and 4.

Although the referees are correct to point out that some properties of layer 4 SST neurons differ from SST neurons in other layers in barrel cortex, recent work finds a clear difference between mouse V1 and S1 – in particular that the vast majority of layer 4 SST neurons in V1 are Martinotti cells (Scala et al., 2019, now stated and cited in the second paragraph of the main text). Furthermore, all of the L4 SST neurons in our study prefer high contrast and have weak direction selectivity, suggesting that these properties are very likely to apply to the subset of L4 SST neurons that are Martinotti cells. Finally, we have also observed in previous work (de Vries et al., 2020) robust reliable responses to high contrast full-field drifting gratings for SST neurons in both layers 4 and 5, suggesting that at least this aspect of SST neuron tuning in layer 4 is not different from layer 5.

9) GeneralizabilityDue to the focus on very specific stimuli, the scope of the study is limited and the results do maybe not generalize beyond large drifting gratings. For instance, it is unclear whether these findings will hold up for smaller stimuli. Other stimulus parameters are not explored systematically (spatial frequency or temporal frequency). Stimulus size is ignored, even though stimulus size could have dramatic effects on the findings presented here. The authors need to discuss these limitations in the manuscript. In particular they should comment on the issue of size tuning, which is missing in the manuscript. Previous studies have shown a very clear size dependence of VIP and SSt neurons (Adesnik and Scanziani, 2012; DiPoppa et al., 2018). The contrast dependence of the VIP and SSt neurons is likely strongly dependent on stimulus size. For instance VIP neurons have small receptive fields (DiPoppa et al., 2018). Because surround modulation is contrast dependent, suppression of PC L2/3 cell activity and VIP activity likely depends on the stimulus size. The authors to discuss these limitations.

Although we have not explored spatial or temporal frequency in this study, we note that suppression of VIP neurons by high-contrast large gratings has been observed across spatial and temporal frequencies (de Vries et al., 2020). For size tuning, we explicitly address this question in the final paragraph with the sentences, “Measurements of size tuning have shown that SST neurons prefer large gratings, suggestive of a role mediating surround suppression, whereas VIP neurons only respond to gratings smaller than those that drive SST neurons. This complementary size tuning parallels the complementary contrast tuning observed here, suggesting that VIP and SST neurons in V1 are tuned for weak and strong inputs, respectively, across multiple stimulus dimensions.” Following those sentences, we also now point to our previous finding that VIP neuron receptive fields as measured with sparse noise stimuli are *largerthan* those of SST and pyramidal neurons (de Vries et al., 2020). This result shows that the relatively small size tuning of VIP neurons does not arise simply from having small linear receptive fields; instead, we believe that VIP neurons have small size tuning due to the weaker stimulus energy of small versus large features. We certainly agree with the reviewers’ point that both size and contrast contribute to VIP and SST responses. Our discussion acknowledges this point and goes further to propose that the parsimonious explanation is that both size and contrast determine the *strength of input* to the cortical circuit which is the determining factor.

10) Interpretation of SSt direction tuning and VIP suppressionThe authors should discuss the interpretation of some of the main effects. In particular, reviewers wondered why VIP neurons are suppressed at high contrast?

We have added a discussion of this point in the seventh paragraph of the main text. Given that VIP neurons have the highest spontaneous activity among the neuron types sampled here (Figure 3; see also Extended Data Figure 1 in de Vries et al., 2020), one possibility is that VIP neurons simply have non-zero spontaneous activity whereas the other neuron types are already close to zero making suppression impossible. Another, related possibility is that suppression of other neuron types *occurs but goes undetected* because both the suppressed and non-suppressed activity levels are below the detection threshold of calcium imaging. Functionally, the high spontaneous activity of VIP neurons enables the cortical circuit to raise or lower the amount of disinhibition of pyramidal neurons depending on stimulus contrast. Our SSN modeling results demonstrate that suppression of VIP neurons below baseline helps the network to maintain stability in response to strong external inputs.

They also wondered why the response of SST neurons at low contrast is not direction modulated – given that the VIP neurons have strong direction tuning?

This is a very interesting question. Note that the SST neurons have a bias toward zero degrees at contrasts greater than 20% whereas SST neurons have weak or no response to any direction at contrasts less than or equal to 20%. One possibility is that SST neurons would also respond to zero degrees at low contrast in the absence of inhibition from VIP neurons. Another possibility is that the responses of SST neurons are simply too small at low contrast to detect direction tuning.

Do VIP interneurons indeed suppress the SSt neurons recorded here? This assumption is being made but it is never tested or argued for these specific laminar recordings from the literature.

Layer 4 SST neurons in V1 are mostly, but not entirely, Martinotti cells (Scala et al., 2019). Although we cannot be certain which of the SST neurons that we recorded are Martinotti cells, the tuning curves of all of the SST neurons look very similar suggesting that our findings apply to Martinotti cells and possibly also non-Martinotti SST neurons. We have added a statement to the main text clarifying this distinction between subtypes of SST neurons and motivating the imaging of Sst mice in layer 4.

11) Optic FlowThe interpretation of optic flow appears to be problematic and needs to be discussed. The authors interpret their data in terms of optic flow related to locomotion. However, the peak of VIP neurons occurs at 45 degrees rather than 0 degrees, but the authors write that 0 degrees corresponds to front-to-back motion (main text, second paragraph). Does the interpretation of the authors make sense given this discrepancy?

We do not believe that the discrepancy is large enough to warrant an interpretation that the mice perceive the direction of motion to be substantially different from front-to-back. First, the peak of VIP neurons is fairly centered between 0 and 45 degrees (new Figure 3D), suggesting a more modest discrepancy of 20 to 25 degrees. Second, our imaging experiments were performed on head-fixed mice that were standing on a wheel that permits them to run in place. The running wheel is angled slightly upward a few degrees to facilitate running during head-fixation which might also influence the mouse’s perception of egocentric angle of visual motion.

12) Modelling:There are major concerns about parameter and model selection. Several findings hard to account for with the model. The description of the model needs to be substantially improved.

These points are addressed where they are raised in detail below.

The tuning of VIP cells in 3b is much wider than that observed in the data (1b), and this could be a major problem for the model. The authors should explain this.

Figure 5B (previously Figure 3B) shows the tuning of pyramidal to VIP connections. Perhaps this comment was intended to refer to Figure 5C (previously Figure 3C) which shows the direction (and contrast) tuning of VIP neuron activity. The tuning of the VIP responses in the SSN shown in Figure 5C (previously Figure 3C) are on the order of 10 to 20 degrees, which is not much wider than the tuning of VIP neurons in mouse V1 shown in Figure 2 (previously Figure 1). Our experiments measured direction responses at increments of 45 degrees and we do not claim that the width of direction tuning for VIP neurons is substantially narrower than this sampling permits us to discern.

The authors should make a better effort to explain the model better in the text. For the modelling, authors need to provide more explanation on why SSNs were selected, and how they work etc. In the current version of the paper, it is necessary to refer to the very difficult Rubin et al., 2015 paper to understand what the model does.

The revised manuscript has a more thorough motivation and introduction to SSNs. In brief, SSNs were selected because a few universal features of cortical circuits (e.g. recurrent excitation, feedback inhibition, and supralinear f-I curves) can account for a wide variety of contrast-dependent phenomenology (Rubin et al., 2015) in addition to a variety of other phenomenology that we now state and cite.

Also, reviewers would like to see that the canonical results from Rubin et al. still work in the revised model, and commented that this is an essential addition to the paper.

In addition to further exploration of network behavior across model parameters (see next point), we have added a section (main text) and Figure 5G-L on the analysis of the inhibitory stabilization behavior of the network as a function of the model parameters. Specifically, we performed linear stability analysis of the excitatory portion of the network (by computing the eigenvalues of the excitatory-to-excitatory submatrix of the Jacobian for the linearized network dynamics; full details in the Materials and methods section) to determine whether or not the steady state firing rates of the network would be stable in the absence of feedback inhibition. In other words, this analysis shows whether the network is inhibitory stabilized or not inhibitory stabilized. It is possible that achieving a high gain makes the recurrent excitation unstable, requiring strong inhibitory stabilization to prevent runaway excitation. Surprisingly, we found that high gain at low contrast effect due to VIP neurons does not require the network to be in an inhibitory stabilized regime.

The authors should include a discussion of the process of parameter selection for the model (how much fine tuning was required, what happens when connection weights deviate somewhat etc). In general, the scope of the model to be somewhat limited given that model parameters are not systematically explored or fitted based on the data.

The revised manuscript includes a wider exploration of model parameters which has yielded additional insights into the behavior of the model. In particular, we have examined the network dynamics as a function of two crucial parameters: 1) the strength of the VIP to SST connection (Figure 5G-I and main text) and 2) the relative amount of inhibition that pyramidal neurons receive from PV versus SST neurons (Figure 5J-L and main text). We show in Figure 5 (previously Figure 3) that our main findings (specifically, increased gain at low contrast due to VIP neurons) hold across a wide range of values for these parameters as well as how the magnitude of the gain effect varies with these parameters.

Further, the model makes strong assumptions on the inputs that these different neuron types presumably receive from external drive. These limitations need to be discussed.

Correct, our model assumes that pyramidal and PV neurons receive direct external (e.g. thalamic) input whereas VIP and SST neurons do not. The lack of direct thalamocortical input to SST neurons (and confirmation of direct thalamocortical input to pyramidal and fast-spiking putative PV neurons) has been observed in mouse somatosensory cortex (Cruikshank et al., 2010). While we also assume that VIP neurons do not receive direct external input, the responses of VIP neurons to weak stimuli would be expected to increase if this additional excitatory input to VIP neurons were added to the model; therefore, our assumption of no direct external input to VIP neurons is conservative in relation to our main conclusion. We have now elaborated on this reasoning in the eleventh paragraph of the main text.

The main text to be improved with a more in-depth explanation for using SSNs (stabilized supra-linear network).

Again, the revised manuscript has a more thorough motivation and introduction to SSNs.

Finally, rectified quadratic response functions for neurons are not common in RNNs, and should be motivated.

Rectified quadratic single neuron transfer functions have been used in previous studies of SSNs (Ahmadian et al., 2013, Rubin et al., 2015) for simplicity of mathematical analysis (e.g. the derivative of x^2^ is simply 2x) while still having the supralinear property. We use the same single neuron transfer function to maintain consistency with the existing SSN literature.

Also, no explanation/citation is provided for the choice of parameters (input, connections) for PV neurons.

We thank the reviewer’s for pointing out that a citation was not provided. We have now cited work that found the orientation tuning of PV neurons to be broad (Kerlin et al., 2010) in the eleventh paragraph of the main text.

With respect to the conclusion – "These results indicate that Vip-mediated disinhibition is capable of producing substantial increases in gain at low contrast despite low activity of the intermediate SST neuron population." The authors should explain why this is not a circular argument and a strange way to summarize the results of the model. Reviewers pointed out that it is clear from Figure 3D that the suppression of Sst activity by Vip neurons at low contrast is what enables supra-linear responses of Cux2 neurons at low contrast.

SST neurons have very weak responses to low contrast gratings. One might (mistakenly) conclude that even full suppression of a small amount of SST activity can only result in a small increase in pyramidal neuron activity. Our intention with this sentence was to emphasize that, due to recurrent excitation and supralinear single neuron transfer functions, the suppression of a relatively *small* amount of SST neuron activity can drive *large* increases in pyramidal neuron activity. We have now elaborated this reasoning in the tenth and twelfth paragraphs of the main text.

The bias of direction tuning in L2/3 neurons is very weak. How can the authors account for the contrast enhancement of these neurons in their VIP model? Should this enhancement not be extremely specific to the 45 degrees angle?

Approximately half of the L2/3 pyramidal neurons are low contrast preferring (Figure 2E) whereas the over-representation of 0 and 45 degrees-preferring L2/3 pyramidal neurons is relatively small. Our measurements of VIP neuron contrast and direction tuning is consistent with the hypothesis that VIP neurons can cause a larger over-representation based on (low) contrast preference than (front-to-back) direction preference. Indeed, VIP neuron activity is higher at low contrast than high contrast across all directions (i.e. the suppressed by high contrast effect) *andVIP neuron activity is even higher* for front-to-back motion than other directions of motion at low contrast (Figure 3D). We now discuss this aspect of the model explicitly in the main text.